# *Legionella* DotM structure reveals a role in effector recruiting to the Type 4B secretion system

Amit Meir[1], David Chetrit[2], Luying Liu[2], Craig R. Roy[2] & Gabriel Waksman[1,3]

*Legionella pneumophila*, a causative agent of pneumonia, utilizes the Type 4B secretion (T4BS) system to translocate over 300 effectors into the host cell during infection. T4BS systems are encoded by a large gene cluster termed *dot/icm*, three components of which, DotL, DotM, and DotN, form the "coupling complex", which serves as a platform for recruitment of effector proteins. One class of effectors includes proteins containing Glu-rich/E-block sequences at their C terminus. However, the protein or region of the coupling complex mediating recruitment of such effectors is unknown. Here we present the crystal structure of DotM. This all alpha-helical structure exhibits patches of positively charged residues. We show that these regions form binding sites for acidic Glu-rich peptides and that mutants targeting these patches are defective in vivo in the translocation of acidic Glu-rich motif-containing effectors. We conclude that DotM forms the interacting surface for recruitment of acidic Glu-rich motif-containing *Legionella* effectors.

[1] Department of Biological Sciences, Institute of Structural and Molecular Biology, Birkbeck, Malet Street, London, WC1E 7HX, UK. [2] Boyer Center for Molecular Medicine, Department of Microbial Pathogenesis, Yale University, 295 Congress Avenue, New Haven, CT 06536-0812, USA. [3] Institute of Structural and Molecular Biology, University College London, Gower Street, London, WC1E 6BT, UK. Correspondence and requests for materials should be addressed to G.W. (email: g.waksman@mail.cryst.bbk.ac.uk)

Legionella spp. are intracellular parasites of protozoa, such as Acanthamoeba spp. and Naegleria spp[1], found in freshwater environments. Legionella pneumophila, the best-studied species of the genus, is also an opportunistic pathogen that is the causative agent of the acute pneumonia known as Legionnaire's disease[2] in humans.

L. pneumophila infects alveolar macrophages, using the host's own mechanism of phagocytosis. The resulting legionella-containing vacuoles (LCVs) escape the endocytic maturation process by resisting fusion to bacteriocidal lysosomes and acidification of the LCV[3]. Over time the LCVs acquire properties similar to the endoplasmic reticulum via fusion with ER-derived vesicles, forming what is known as the replicative phagosome—an organelle in which L. pneumophila establishes its replicative niche[4].

The infection mechanism by which Legionella recruits and manipulates transport of the phagosome involves over 300 bacterial genes encoding different proteins, called effectors, that are secreted from the bacteria to the host cell[5]. Although the biochemical function of most effector proteins remains to be determined it has been shown that these proteins influence a number of host cellular processes, including cell signaling pathways, membrane transport pathways, and host translation.

Legionella utilizes a Type 4 secretion (T4S) system termed "dot/icm"[6–8] to translocate these effector proteins into the host cell[9,10]. T4S systems are comprised of protein complexes that assemble into a machine that traverses the bacterial inner membrane and cell envelope. These systems are usually used to transport DNA molecules from a donor cell to a recipient cell during conjugation, to take up macromolecules from the environment during transformation, or to transport effector proteins from the bacterium to a host cell during infection. The T4S system is composed of a secretion channel and a coupling protein/complex that recruits substrates and delivers them to the secretion channel[11–13]. The coupling protein/complex includes an inner membrane ATPase belonging to the VirD4 family of proteins[14–16].

T4S systems can be divided into two classes: The T4AS system class, which is mostly used for DNA delivery, and the T4BS system class, which is mostly utilized for protein secretion, but has been shown to transport genomic material as well[17]. Although some structural and functional similarities can be found between the two classes[18], the T4BS systems are larger, comprising, for example in L. pneumophilla, 27 components against 12 components only for conjugative T4AS systems. Also, while only one protein, VirD4, mediates coupling between substrate recruitment and delivery to the secretion channel in T4AS systems, a "coupling" complex is needed in T4BS systems.

The T4BS secretion coupling complex of L. pneumophila comprises the inner membrane AAA+ ATPase DotL (IcmO) and two inner membrane/cytoplasmic components, DotM (IcmP) and DotN (IcmJ)[19]. In addition, a complex of two cytoplasmic chaperones, IcmS and IcmW (IcmSW), has been shown to mediate recruitment of a subset of effectors[20,21] (termed "IcmSW-dependent effectors") to DotL, by binding to a recognition sequence on the C terminus of DotL[19,22]. DotL is a homolog of VirD4[23], the coupling ATPase of the T4AS secretion, yet it has an extended C terminus which is ~200 amino acids longer. Recently (while this manuscript was being revised), the crystal structure of the C-terminal tail of DotL bound to IcmSW, that of the C-terminal tail of DotL bound to DotN, and that of the DotM cytoplasmic domain alone have been reported[24]. These structures reveal the molecular basis of DotN- and IcmSW- mediated stabilization of the DotL C-terminal tail, but the function of DotM could not be inferred from the structure[24].

Among effectors that do not depend on IcmSW for binding to the DotMLN coupling complex (termed "IcmSW-independent effectors"), a subset appears to be mediated by a secretion signal sequence rich in Glu residues and located at the C terminus[25–28] (termed thereafter as "Glu-rich motif or E-block"). However, the interacting platform for Glu-rich motif effectors on the DotMLN coupling complex is unknown. In this study, we determine the crystal structure of the cytoplasmic domain of DotM and observe large patches of basic residues, leading us to hypothesize that DotM might form the recruiting platform for Glu-rich motif-containing effectors. This hypothesis is tested using a series of isothermal titration calorimetry experiments in which a number of Glu-rich and more neutral peptides as well as DotM variants mutated in the basic patches are analyzed for binding. Allelic replacement is used to create isogenic dotM mutant strains of L. pneumophila. These dotM mutant strains are characterized using replication assays and effector translocation assays. These experiments identify DotM as a potential binding platform for recruitment of acidic Glu-rich, IcmSW-independent effectors.

## Results

**DotM structure.** The cytoplasmic domain of DotM (residues 119–380, DotM119) was identified based on membrane-spanning region prediction (MemSAT[29]) and cloned into a pET backbone vector, with a hexa-histidine tag followed by an HRV-3C protease cleavage site at its N terminus. DotM119 was expressed in E. coli BL21(DE3) BLR cells (Novagen), but appeared partially truncated at the N terminus at residue 153. Thus, a shorter construct (residues 153–380, referred thereafter as DotM153) was cloned, expressed, and purified (Fig. 1a, b, Supplementary Figure 3). DotM153 (26.5 kDa) was found to be a dimer in solution, according to SEC-MALS studies. However, the longer, less stable construct, DotM119 (31 kDa), eluted from the Superdex200 column as a sharp monomer peak.

DotM119 crystallized as very thin needles that diffracted poorly. DotM153 crystals however diffracted to 1.8 Å resolution (Table 1). These crystals contained two molecules in the asymmetric unit. The DotM153 structure was solved using the single-wavelength anomalous dispersion (SAD) phasing method applied to crystals of the seleno-methionine-substituted protein (see details in Methods). This resulted in an electron density map of excellent quality in which a model could be readily built and refined (Fig. 1c and Table 1).

The DotM153 crystal structure is similar to that published by Kwak et al.[24] (PDB entry code 5 × 1U; root mean square deviation (RMSD) in Cα atoms of 0.4 Å) and reveals an all-α-helical fold, comprising 13 α-helices connected by loops, many of them proline-rich (Figs. 1d, e and 2). Residues 179–188 in monomer A in the loop connecting helices α1 and α2 were not resolved, indicating flexibility in this region of the protein. The RMSD in Cα atoms between the two monomers was 0.17 Å². The pear-shape structure is formed of a narrower end made by two elongated sequences emanating from both the N- and C-terminal regions of the sequence: α1, α2, and part of the α3 on one side and α13 and α12 on the other. The remaining α-helices (α4 to α11) form a larger compact structure with helices α7, α8, α9, and α10 tightly wrapping around a helical bundle formed by α4, α5, and α6 (Fig. 1d, e). Differences between this structure and that published by Kwak et al.[24] include the region formed by residues 176–193, which form a short α-helix (α2) and a short loop in the structure presented here, rather than a long flexible loop in the other. Also, the C-terminal helix, α13, is missing in the structure by Kwak et al.[24].

Submission of the coordinates to the DALI server[30] did not return any known structural homologs and therefore the structure of DotM is representative of an all alpha-helical fold. Thus, the structural homology search did not provide any clue as to what the function of DotM might be.

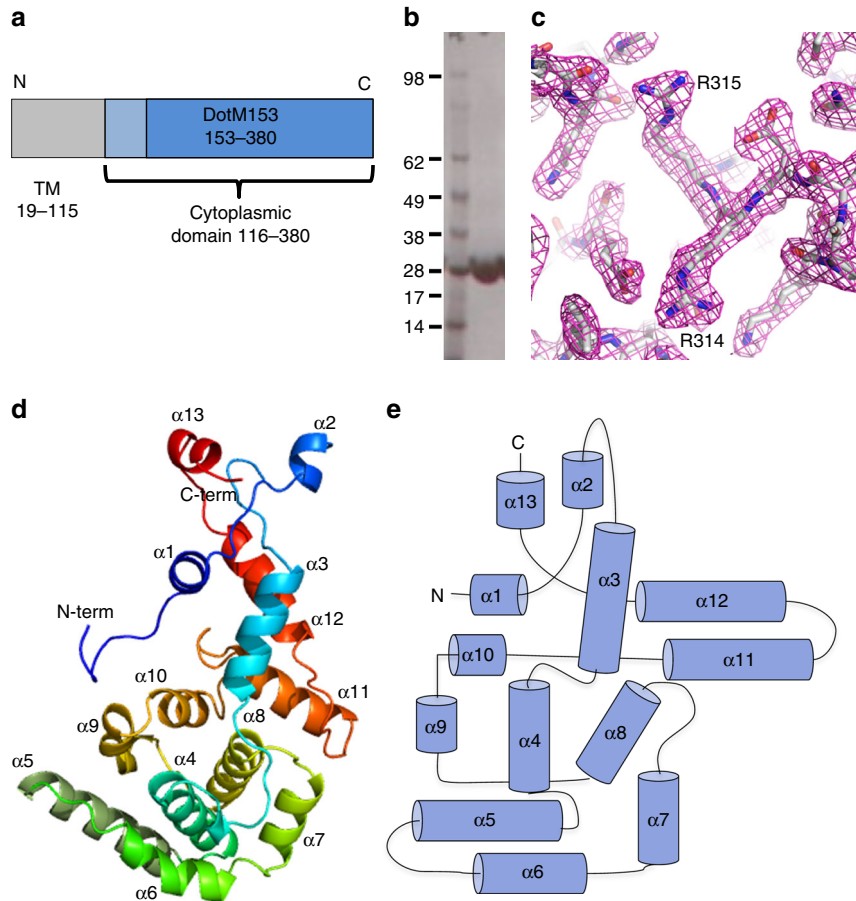

**Fig. 1** DotM153 characterization and structure. **a** Domain organization of DotM. Residues 19–115 are predicted to form three transmembrane helices. The cytoplasmic domain (116–380) and the stable DotM153 fragment are shown in light and dark blue, respectively. **b** Purification of DotM153. A single band of DotM153 was observed in the SDS-PAGE analysis of the purified protein. Molecular weight markers (kDa) are shown in lane at left. Full gel figures are available in Supplementary Figure 3. **c** Experimental SAD electron density map of the DotM153 calculated at a resolution of 2.15 Å and contoured at a 2.0σ level. The map is in wire representation colored in magenta. The refined model built into the electron density is shown in stick representation, color-coded by atom type with C, O, and N atoms in gray, red and blue, respectively. Residues Arg314 and Arg315, participating in effector binding, are labeled. **d** Cartoon diagram representation of the crystal structure of the DotM153 monomer. The structure is rainbow colored and its 13 helices are labeled α1–13. The N and C termini are indicated. **e** Topology diagram of DotM153. Helices are shown as cylinders and labeled α1–13. N- and C-termini are indicated

DotM sequences are highly conserved among *Legionella* species, and DotM proteins are present in *Fluoribacter dumoffii*, *Rickettsiella grylli* and *Coxiella burnetii* with 77, 43 and 38% identity, respectively. Thus, there is likely structural similarity between the DotM proteins encoded in all these species (Fig. 2). DotM homologs encoded by other T4BS system proteins, such as the TrbA protein in the R64 conjugation plasmid, are more divergent, and in T4AS systems DotM shows a mild homology with TrbA of the IncI1 conjugative system[31] (Fig. 2). TrbA function is unknown and thus, here again, no functional insight could be derived from the structure-based sequence alignment.

**DotM binds acidic Glu-rich sequences**. While the structural homology search and structure-based sequence alignment yielded no clue as to the function of DotM, functional implications could however be derived from a striking feature of the DotM153 structure: its highly charged surface. Indeed, DotM153 contains 33 surface-exposed Lys/Arg and 28 surface-exposed Glu/Asp residues scattered around the entire surface of the protein (Fig. 3). This represents 26% of the total number of residues in DotM153, 31% of its surface-exposed residues. Importantly, some of these charged patches are conserved among DotM homologs (Fig. 2).

Because a subset of *L. pneumophila* effectors contains a negatively charged Glu-rich signal motif at their C terminus, we hypothesized that DotM153 may be involved in effector binding. Hence, several synthetic peptides, 27–30 amino-acid long, derived from the C-termini of *L. pneumophila* effectors and based on the work published by Lifshitz et al.[27] were selected for affinity binding assays using Isothermal Titration Calorimetry (ITC). Six peptides were chosen based on the hidden semi-Markov model ranking reported by Lifshitz et al. (Table 2). Three of these peptides (derived from effectors CegC3, Lpg1663, and Lem8) were highly ranked, and showed IcmSW-independent secretion mechanism[27]. Two effectors (derived from Lem21 and LegC8) had a relatively low rank and were shown to be IcmSW dependent. Finally, a peptide termed "OSM" (Table 2) was chosen: the sequence of OSM is a single-residue substitution (L to E at the very C terminus) of the OSS peptide described in Lifshitz et al.[27] as a computationally derived consensus sequence for Glu-rich motifs. This single mutation, known to abrogate effector transport in vivo, was required as the OSS peptide itself is not soluble and therefore cannot be used in ITC experiments. These peptides were subjected to ITC to assess their binding to DotM153 (Fig. 4 and Table 3). Peptides CegC3, Lpg1663, and OSM that contain a Glu-rich motif exhibited high affinity to

**Table 1 X-ray data collection and refinement statistics**

|  | DotM153 SeMet | DotM153 Native | DotM153 R196E/197E | DotM153 R217E | DotM153 R314E/315E |
|---|---|---|---|---|---|
| *Data collection* | | | | | |
| Space group | P6$_5$ | P6$_5$ | P6$_5$ | P6$_5$ | P1 |
| *Cell dimensions* | | | | | |
| *a, b, c* (Å) | 118.9, 118.9, 66.7 | 118.5, 118.5, 66.3 | 118.4, 118.4, 66.47 | 118.2, 118.2, 66.5 | 46.4, 50.6, 55.4 |
| *α, β, γ* (°) | 90.00, 90.00, 120.00 | 90.00, 90.00, 120.00 | 90.00, 90.00, 120.00 | 90.00, 90.00, 120.00 | 102.5, 97.6, 95.5 |
| Resolution (Å) | 50–2.16 (2.22–2.16) | 102–1.84 (1.89–1.84) | 66–1.8 (1.82–1.79) | 102–2.0 (2.16–2.0) | 53–2.1 (2.2–2.1) |
| $R_{sym}$ or $R_{merge}$ | 0.253 (1.137) | 0.098 (0.029) | 0.168 (0.046) | 0.181 (2.44) | 0.133 (0.5) |
| $I/\sigma I$ | 9.11 (1.5) | 13.9 (1.2) | 12 (2.6) | 7.88 (1.52) | 5.7 (1.3) |
| Completeness (%) | 99.54 (99.2) | 99.85 (100) | 100 (100) | 99.65 (99.2) | 97.1 (95.3) |
| Redundancy | 13.2 (12.6) | 10.1 (9.3) | 6.6 (6.5) | 6.7 (6.8) | 1.8 (1.7) |
| *Refinement* | | | | | |
| Resolution (Å) | 50–2.16 | 102–1.8 | 66–1.8 | 102–2.0 | 53–2.1 |
| No. of reflections | 386,229 | 474,663 | 331,246 | 204,508 | 45,084 |
| $R_{work}/R_{free}$ | 19.1/22.7 | 20.1/25.0 | 20.2/25.4 | 22.4/26.4 | 19.3/25.4 |
| *No. of atoms* | | | | | |
| Protein | 3516 | 3525 | 3561 | 3472 | 3385 |
| Ligand/ion | 0 | 12 | 20 | 12 | 0 |
| Water | 131 | 229 | 344 | 128 | 124 |
| *B-factors* | | | | | |
| Protein | 31.25 | 45.95 | 34.76 | 38.80 | 58.47 |
| Ligand/ion | NA | 55.06 | 75.8 | 21.45 | NA |
| Water | 30.72 | 47.64 | 40.73 | 34.11 | 58.83 |
| *Root mean square deviations* | | | | | |
| Bond lengths (Å) | 0.018 | 0.019 | 0.019 | 0.017 | 0.0146 |
| Bond angles (°) | 2.08 | 2.01 | 2.03 | 1.91 | 1.800 |

Highest resolution shell is shown in parenthesis

DotM153 ($K_D$ values of 0.19, 0.7, and 0.35 µM, respectively), whereas Lem21 and LegC8 which do not contain C-terminal Glu-rich motifs did not bind. Lem8, which received a high score according to statistical calculation by Lifshitz et al., did not bind either; its sequence contains a large number of Glu residues but its overall pI is 6.3 because Glu residues are neutralized by an equal number of adjacent Lys residues. Thus, DotM is able to bind Glu-rich peptides with high affinity, provided that the overall pI of these peptides is low: we termed this subset of Glu-rich peptides "acidic Glu-rich motifs/peptides". Thus, DotM might be involved in recruitment of a set of effectors characterized by highly negatively charged (acidic) Glu-rich C-terminal motifs.

Further binding studies were conducted with several DotM mutants, where Arg residues (single or double) on the protein's surface were mutated to Glu. Five mutants were produced (Fig. 3): R196E/R197E (termed "M1"), R314E/R315E (termed "M2"), R347E/R348E (termed "M3"), R217E (termed "M4"), and R262E (termed "M5"). These residues were chosen for amino-acid substitution because they reside in the most highly charged surface of the structure (see location of the targeted residues in Fig. 3) and probe widely the various regions of this surface. Binding to these DotM153 variants was assayed using ITC against the CegC3 peptide, the peptide exhibiting the highest affinity for the wild-type DotM protein (Fig. 4 and Table 3). Binding was abrogated in M1, M2, and M4, indicating that these Arg patches are involved in binding of acidic Glu-rich peptides. M3 and M5 were unaffected suggesting that they are not involved in Glu-rich sequence binding. The Lpg1663 and the consensus OSM peptide was also assessed for binding to the M1 and M2 mutants and binding was found to be also abrogated (Fig. 4 and Table 3; Lpg1663- or OSM-binding to other mutants was not tested because wild-type DotM, M1 and M2 display the same binding behavior for the Lpg1663 and OSM peptides as for CegC3). To ascertain that the observed binding deficiency in M1, M2, and M4 variant proteins was not due to folding defects caused by the mutations, these three proteins were crystallized and their structures were determined and shown to be virtually identical to the wild-type protein (Supplementary Figure 1). Thus, the mutations do not introduce structural defects, suggesting that the ITC binding results reflect an effective role of the residues in binding.

From the experiments described above, we conclude that acidic Glu-rich peptides bind to DotM and that at least three sites (R196E/R197E, R314E/R315E, and R217E) are important for binding, suggesting that acidic Glu-rich peptides bind in an extended manner along the entire length of DotM. Interestingly, when a shorter, 10 amino-acid-long peptide derived from the CegC3 peptide (see sequence in Table 2) was tested for binding to the wild-type DotM153 protein, no binding was observed (Fig. 4 and Table 3), indicating that not only the entire DotM binding surface is required for binding, but also the entire peptide surface.

The biochemical, structural, and mutational data described above were next used to produce an in silico model of CegC3-interaction with DotM153, using the docking server CABS[32], and subsequently refining the resulting model using FlexPep-Dock[33,34], a ROSETTA-based server. The final model suggests that CegC3 is primarily helical. Binding is observed between the main middle Glu-rich helical motif of CegC3 (residues E155, E158, and E159; see numbering in Table 2 and interactions in Fig. 5) to the "central patch" residues R314 and R315, as well as between R217 of DotM to peptide residue E162 (Fig. 5). Other interactions are between residues R196 and R197 of DotM and CegC3 E145 and a stacking interaction between these two residues and the side chain of CegC3 F146 (Fig. 5). To validate this model, E145 was mutated to Ala and, using ITC, this mutant CegC3 peptide was shown to no longer bind to DotM (Fig. 4). The observation that a single mutation abrogates binding suggests that positioning of Glu residues along the surface of DotM might be important and therefore that sequence specificity might play a role in acidic Glu-rich motif interactions with DotM.

**DotM mutants display effector translocation defects**. To determine whether the residues in DotM important for effector binding in vitro were required for Dot/Icm function in vivo, the M1, M2, and M4 substitution mutations were introduced in the *L. pneumophila* (Lp01) chromosome by allelic exchange. Two were

successfully obtained (M1 and M4) but attempts at introducing M2 failed for reasons that remain unclear but could be due to a destablization of the DotMLN coupling complex resulting in activation of processes that are stressful to the cell. Production of the DotM, M1 and M4 proteins in cells was monitored using anti-

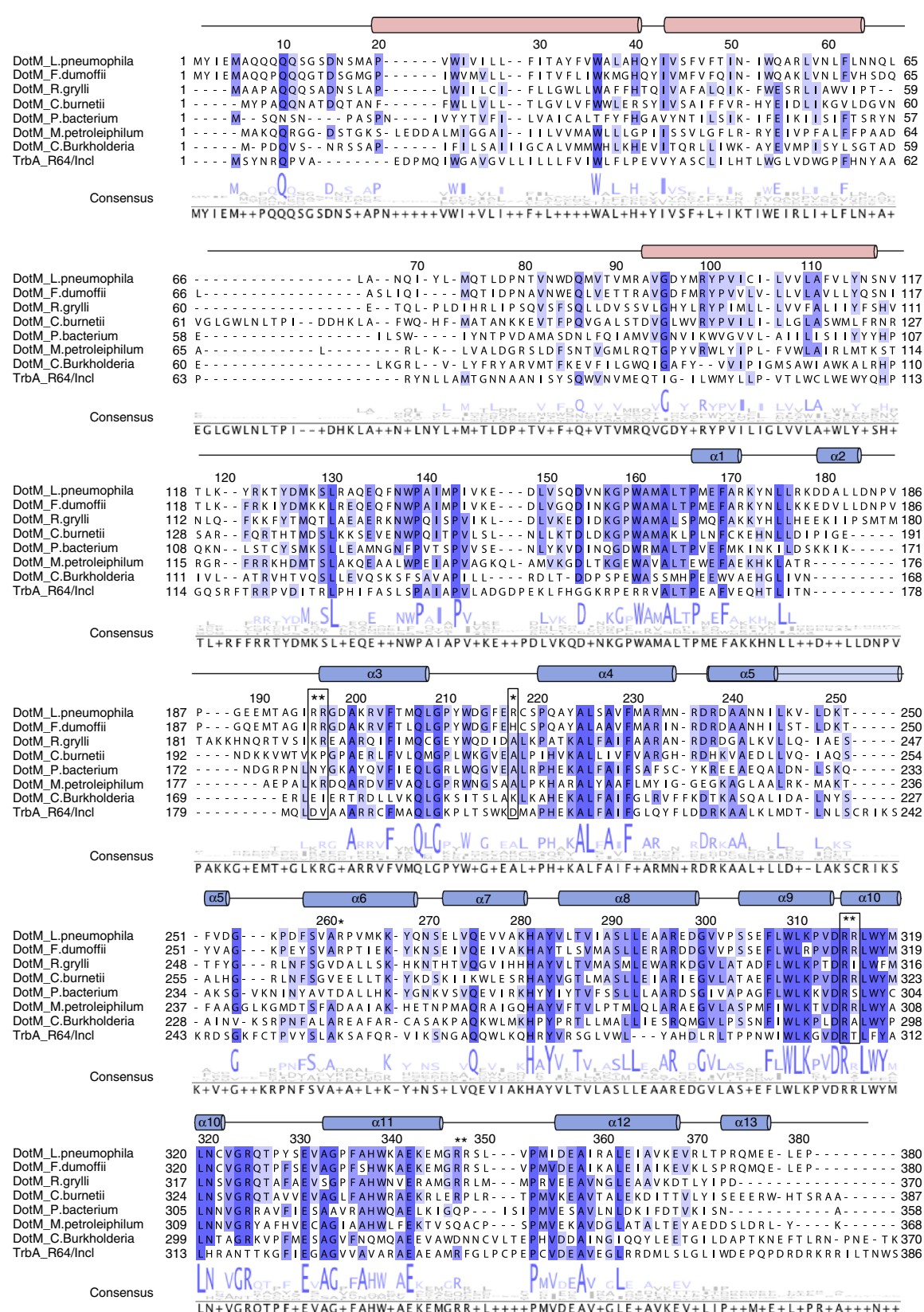

DotM antibodies and all wild-type and variant DotM proteins were shown to be produced in equal quantities (Supplementary Figures 2A and 3).

The effect of the DotM mutations on *L. pneumophila* intracellular replication in eukaryotic hosts was tested using both the protozoan host *Acanthamoeba castellanii* and the mammalian J774A.1 murine macrophage cell line. Levels of intracellular replication displayed by the *dotM* mutant strains were compared to the wild-type *L. pneumophila* (Lp01) strain and a "ΔT4BS" strain[35], which carries chromosomal deletions of three loci: *icmX–dotA*, *dotB–dotD*, and *icmT–dotU* (the later locus includes the *dotL* and *dotM* genes). Remarkably (given that acidic Glu-rich motif-containing effectors represent only a subset of *Legionella* effectors), the M1 and M4 mutations resulted in a significant decrease in *L. pneumophila* replication in the macrophages (up to 70–80%; Fig. 6b, left panel). However, in *A. castellanii*, M1 reduces intracellular growth by up to 90% (after 24 h) while the slight reduction in intracellular growth of the M4 mutant is not statistically significant (Fig. 6b, right panel). Thus, defects in intracellular replication displayed by the *dotM* mutant strains were more apparent in macrophages than in the natural host *A. castellanii* (Fig. 6a, b). Intriguingly, M4 appears to inhibit growth much more markedly in macrophages, perhaps indicating differential effects of acidic Glu-rich motif-containing effectors depending on the host.

The significant decrease in intracellular growth by *Legionella dotM* mutants, particularly in macrophages, is likely due to reduce ability of *dotM* mutants to export acidic Glu-rich motif-containing effectors. To test this hypothesis, the ability of *dotM* mutants to translocate effector proteins was next tested. The translocation reporter consisted of the calmodulin-dependent adenylate cyclase domain (Cya) of the *Bordetella pertussis* adenylate cyclase toxin[25] to which the C terminal 30-residue sequence (which includes the secretion signal) of several *L. pneumophila* effectors (termed herein Cter) was fused to the C terminus of Cya. The fused secretion signals used in this part of the study were from the same effectors from which the synthetic peptides tested in the ITC experiments were derived, i.e., CegC3, Lpg1663, Lem21, and LegC8 (yielding the reporter fusions named thereafter $Cya\text{-}CegC3_{Cter}$, $Cya\text{-}Lpg1663_{Cter}$, $Cya\text{-}Lem21_{Cter}$, and $Cya\text{-}LegC8_{Cter}$, respectively). In addition, the sequence corresponding to the OSS peptide was also fused to the C terminus of Cya (named $Cya\text{-}OSS_{Cter}$). Thus, the C-terminal sequences of two IcmSW-independent acidic Glu-rich motif-containing effectors (CegC3 and Lpg1663), two IcmSW-dependent effectors with no Glu-rich motifs (Lem21 and LegC8), and the consensus OSS sequence were tested for translocation of Cya in vivo using the wild-type, M1, and M4 *Legionella* strains. We observed that the M1 and M4 *dotM* mutants producing the reporter fusions $Cya\text{-}CegC3_{Cter}$, $Cya\text{-}OSS_{Cter}$, or $Cya\text{-}Lpg1663_{Cter}$ displayed lower levels of effector translocation compared to the wild-type strain of *L. pneumophila* ($Cya\text{-}CegC3_{Cter}$: 50% and 30% decrease compared to wild-type DotM for M1 and M4, respectively; $Cya\text{-}OSS_{Cter}$: 50% and 40% decrease compared to wild-type DotM for M1 and M4, respectively; $Cya\text{-}Lpg1663_{Cter}$: 60% and 55% decrease compared to wild-type DotM for M1 and M4, respectively (Fig. 6c)). The $Cya\text{-}LegC8_{Cter}$ and $Cya\text{-}Lem21_{Cter}$

reporter fusions were translocated very weakly by all strains (Fig. 6c) and there were no statistically significant differences between strains. Weak translocation of these fusions is presumably due to the absence of internal IcmSW binding sites. To ensure that this is the case, full-length LegC8 was fused to the C terminus of Cya (yielding a fusion termed Cya-LegC8) and translocation of Cya-LegC8 was monitored using the wild-type, M1 and M4 *Legionella* strains (Supplementary Figure 2B). We observed higher levels of translocation, yet still no statistically significant differences between wild-type and mutant strains, suggesting that weak translocation of $Cya\text{-}LegC8_{Cter}$ is indeed due to the absence of an IcmSW-dependent translocation signal. Also, the fact that the full-length Cya-LegC8 fusion is translocated with the same efficiency in all *Legionella* strains indicates that the M1 and M4 mutations do not affect the IcmSW-dependent translocation function of the DotMLN complex. We can therefore safely conclude that DotM and the surface of DotM we identified as binding acidic Glu-rich motif-containing peptides are recruitment platforms for acidic Glu-rich motif-containing effector proteins in vivo.

## Discussion

In this study, we investigate the role of DotM, until now an enigmatic member of the T4BS secretion coupling complex. DotM was considered to have only a structural role in this complex as its deletion results in DotL destabilization[19,23,36]. Until now, only IcmSW, the cytoplasmic complex of the two adaptor proteins, was shown to be capable of recruiting a subgroup of effectors and introducing them to the T4BS coupling protein DotL. Our results suggest that DotM's cytoplasmic domain is capable of binding directly another subgroup of *Legionella* effectors that contain acidic Glu-rich motifs at their C terminus. Although many effectors may use either DotM or IcmSW as recruitment platforms, the use of these platforms might not be mutually exclusive and it is possible that some effectors might use both simultaneously.

Both DotM and IcmSW are unique to the T4BS systems family. However, while IcmSW can be found only in a subgroup of T4BS systems (in *Legionella* species, *C. burnetii* and *R. Grylli*), DotM is found throughout the T4BS system family, and its gene is upstream to its coupling partner ATPase DotL. Even in strains and species that produce the IcmSW complex, many effectors do not depend on it for secretion. Our results suggest that DotM might target a subset of IcmSW-independent effectors characterized by acidic Glu-rich C-terminal sequences. Intriguingly, among the residues we show here to be involved in this recognition process, only R314 is highly conserved, whereas residues R196, R197, and R217, despite their importance in Glu-rich peptide binding, exhibit a lower conservation. What would be the role of DotM in the species where these residues are not conserved? To answer this question, it is relevant to note that, among bacterial species that have T4BS systems, the presence of Glu-rich sequences in effectors' C-termini correlates strongly with the presence of Arg/Lys residues at the effector-binding interface of DotM. Thus, it could be that, in other species that do not have effectors with acidic Glu-rich C-terminal ends, DotM might still

**Fig. 2** Structure-based sequence alignment of the DotM family of proteins. The sequence of DotM from *L. pneumophila* is aligned to its closest homologs from *Fluoribacter dumoffii*, *Rickettsia grylli*, and *Coxiella burnetii* (77%, 43%, and 38% identity, respectively), as well as other DotM homologs from *Piscirickettsiaceae bacterium*, *Methylibium petroleiphilum*, and *Candidatus Burkholderia*. TrbA from R64/Incl conjugation plasmid is also aligned, although its identity is much lower (23%). Highly conserved, strongly conserved, and conserved residues are indicated by dark blue, purple, and lavender, respectively. Residues mutated at this study indicated by an asterisk, and those found to participate in effector binding in *L. pneumophila* are boxed. The secondary structure elements are indicated as observed in the crystal structure (blue cylinders) or predicted transmembrane helices (red cylinders). Secondary structure elements and amino acid numbering at the top of the sequence refer to DotM from *L. pneumophila*

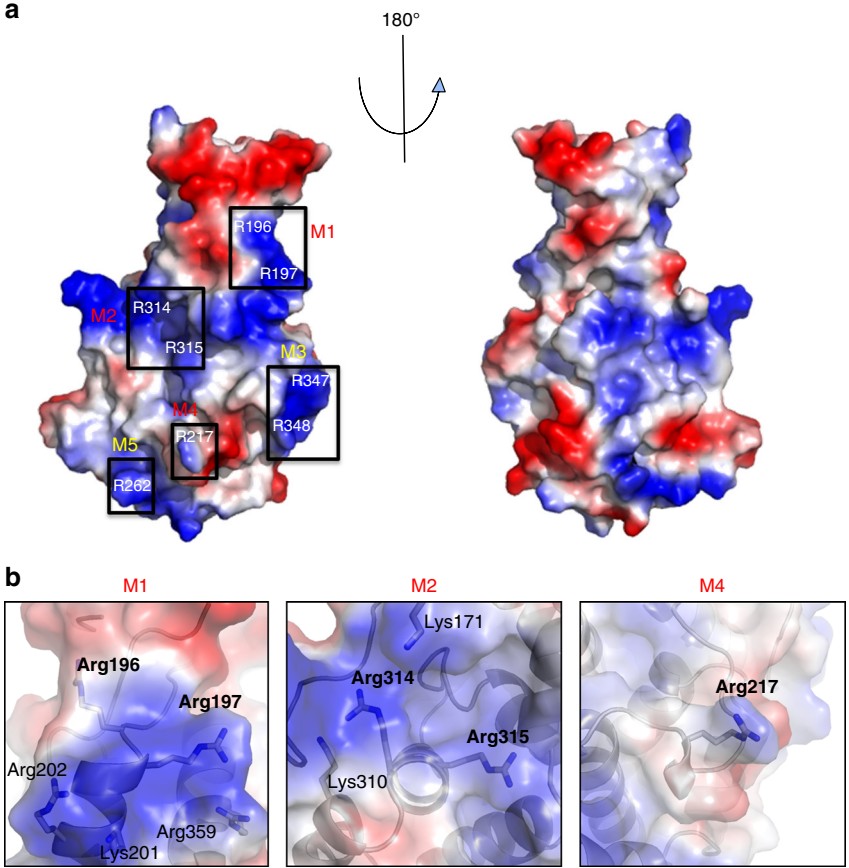

**Fig. 3** Charged residues on the surface of DotM. **a** Electrostatic surface representation of DotM153. Blue and red patches indicate positively and negatively charged surfaces, respectively. Black boxes identify Arg-rich surfaces that were mutated in this study. The boxes are labeled M1-5 according to the nomenclature used to name these mutants (see main text and Table 3), and these labels are colored red for mutants abolishing binding or yellow for mutants that do not affect binding. **b** Zoom-in into Arg-rich patches contributing to peptide binding. Semi-transparent surfaces are as in A. Residues are in stick representation, color-coded as in Fig. 1c

| Peptide | Gene | IcmSW | Sequence |
|---|---|---|---|
| CegC3 | lpg1144 | I | GG- AEFSSSES SENEEKEEEN EESSRFTM (144-168) |
| CegC3 Short | lpg1144 | I | GG- EEKEEEN EES (154-163) |
| Lpg1663 | lpg1663 | ND | GG- PKVVSEDK AESEEENEDE ESRNSASV (143-168) |
| Lem8 | lpg1290 | I | GG- EKTEKTE KTEKTENEQS RNTRGFPI (504-528) |
| OSM | — | I | GG- FSSDDVLLEEEEEEEESSLLSSLKE |
| LegC8 | lpg2862 | D | GG- EEVERTQSLR TDGLSWMPSE QARLSK (611-636) |
| Lem21 | lpg2248 | D | GG- R HFTSSLNKLA SVLEVQLFAN RYVP (720-744) |

**Table 2 Peptides used in this study**

The 27–30 aa long peptides are derived from the C terminus of *L. pneumophila* effectors (effector name and gene number are indicated) and were chosen according to their ranking by Lifshitz et al. Amino acids numbering according to their location in effectors' sequences is indicated in brackets. OSM, the consensus sequence peptide, is a synthetic construct, hence does not have a corresponding gene. All peptides have two glycines at their N-terminal to enhance solubility
I/D = IcmSW Independent/Dependent, ND not determined

play a similar recruitment role, having co-evolved a binding interface suited for the binding of these different, more diverse, C-terminal tails.

DotM and IcmSW might not be the only proteins involved in effector binding. DotN also exhibits patches of acidic and basic residues and the high abundance of arginine and lysine residues in its sequence (12.1%) hints that it might participate in effector binding as well[24]. Another possible effector-recruiting member of the Dot/Icm system is IcmT. IcmT is a small inner membrane protein, upstream to IcmS in the T4BS system operon, containing plenty of positively charged residues (pI = 12) exposed to the

cytoplasm (according to prediction). Further experiments are required to investigate whether this protein also participates in recruiting of effectors with a negatively charged C-terminal signal.

Our structural and biological investigations of DotM reveal an important role of DotM in recruitment of *Legionella* effectors of a particular class; our discovery opens avenues of research aiming at determining exactly how many and which *Legionella* effectors might use DotM as a docking platform. These structural studies of DotM's interaction with *Legionella* effectors could guide the design of therapeutic compounds that block effector recruitment. As well as having therapeutic potential to combat *Legionella*

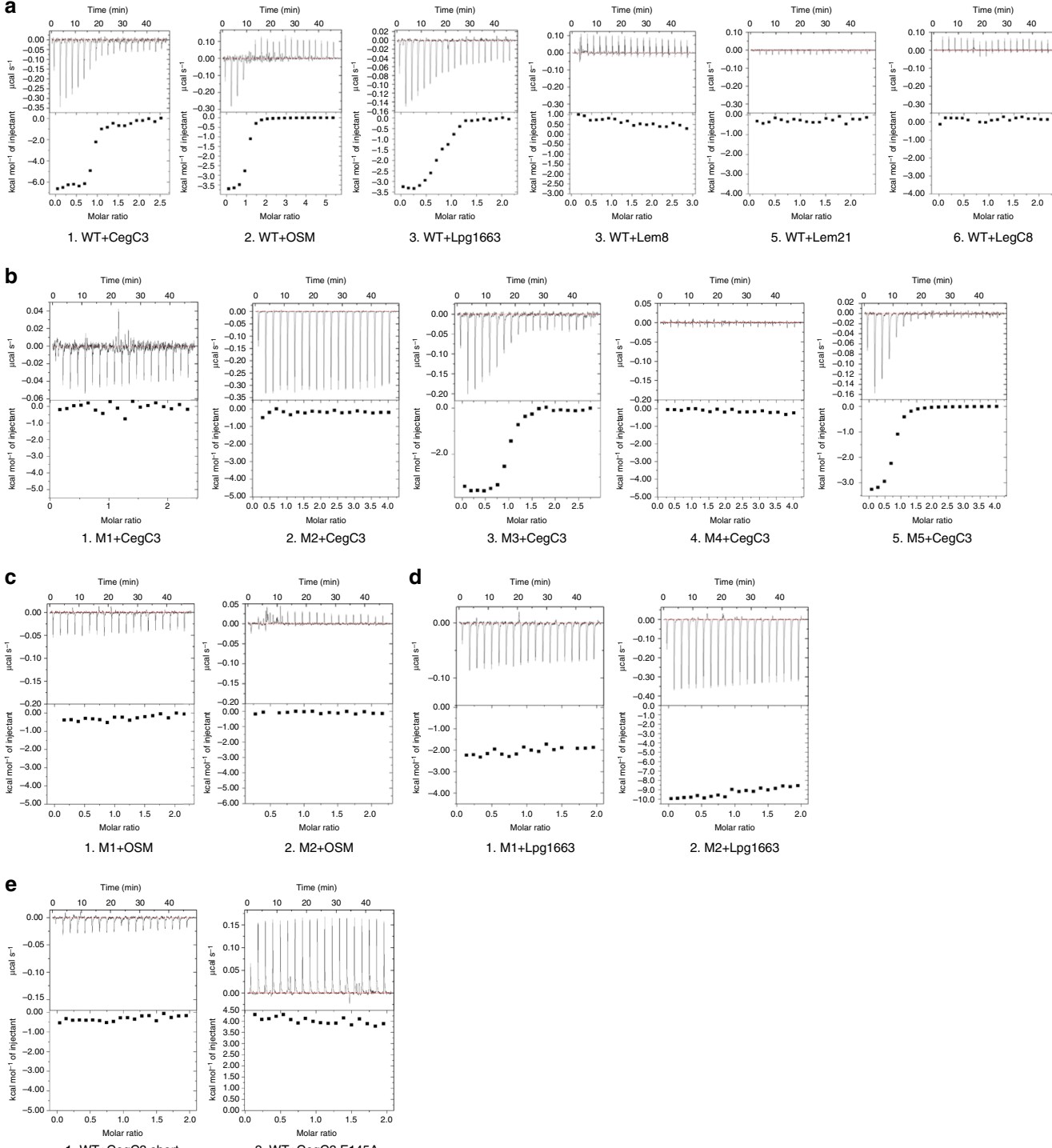

**Fig. 4** Binding of DotM153 and DotM153 variants to various effectors' C-terminal tail peptides. **a** ITC of effector peptide binding to wild-type (WT) DotM153. Glu-rich peptides CegC3, OSM, and Lpg1663 (A1, A2, and A3, respectively) showed high affinity toward DotM, whereas peptides lacking this motif, Lem8, Lem21, and LegC8 (A4, A5, and A6, respectively) did not show any detectable binding. **b** ITC of CegC3 peptide binding to DotM153 mutants (M1–5). While mutants M1, M2, and M4 (B1: R196E/R197E, B2: R314E/R315E, and B4: R217, respectively) abolished binding, mutants 3 and 5 (B3: R347E/R348E and B5: R262, respectively) maintained properties similar to the wild-type DotM153. **c** ITC of OSM peptide binding to DotM153 mutants M1 and M2. The affinity of M1 (C1) and M2 (C2) was measured against the OSM peptide, yielding similar results as CegC3 peptide binding to M1 and M2. **d** ITC of Lpg1663 peptide binding to DotM153 mutants M1 and M2. The affinity of M1 (D1) and M2 (D2) was measured against the Lpg1663 peptide, yielding similar results as CegC3 peptide binding to M1 and M2. **e** ITC of a shorter version of CegC3 (E1) (see Table 2 for sequence) and CegC3 mutant E145A (E2) to wild-type DotM153

**Table 3 Binding dissociation binding constants derived from isothermal titration calorimetry**

| Curve | DotM | Mutated Residues | Peptide | $K_D$ (µM) |
|---|---|---|---|---|
| A.1 | WT | | CegC3 | 0.19 |
| A.2 | WT | | OSM | 0.35 |
| A.3 | WT | | Lpg1663 | 0.70 |
| A.4 | WT | | Lem8 | U.D. |
| A.5 | WT | | Lem21 | U.D. |
| A.6 | WT | | LegC8 | U.D. |
| B.1 | M1 | R196E/R197E | CegC3 | U.D. |
| B.2 | M2 | R314E/R315E | CegC3 | U.D. |
| B.3 | M3 | R347E/R348E | CegC3 | 0.20 |
| B.4 | M4 | R217E | CegC3 | U.D. |
| B.5 | M5 | R262E | CegC3 | 0.40 |
| C.1 | M1 | R196E/R197E | OSM | U.D. |
| C.2 | M2 | R314E/R315E | OSM | U.D. |
| D.1 | M1 | R196E/R197E | Lpg1663 | U.D. |
| D.2 | M2 | R314E/R315E | Lpg1663 | U.D. |
| E.1 | WT | | CegC3 Short | U.D. |
| E.2 | WT | | CegC3 E145A | U.D. |

DotM153 WT and its five mutants (M1–5) were measured for their affinity toward the peptides indicated in Table 2. For each DotM construct, "curve" refers to the corresponding titration shown in Fig. 4
U.D. undetected binding

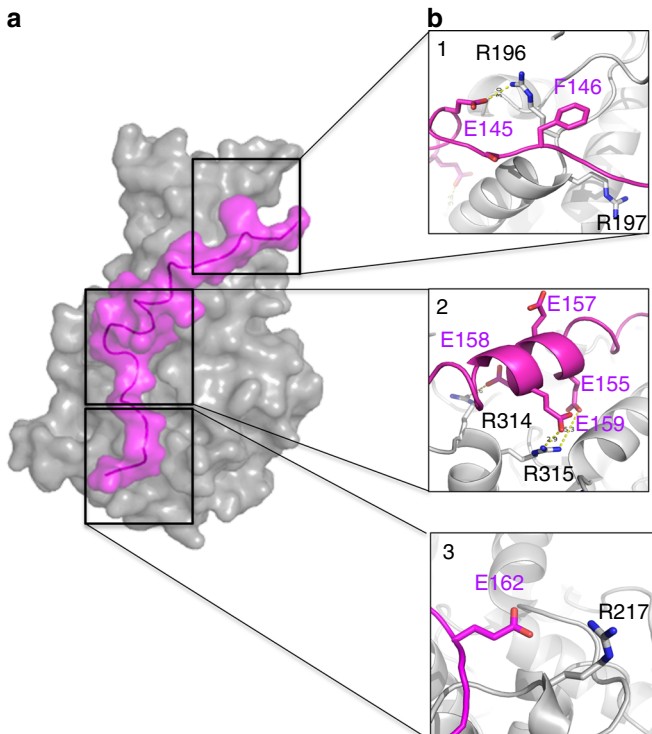

**Fig. 5** Model of CegC3's Glu-rich tail peptide bound to DotM153. **a** Surface representation of wild-type DotM153 (gray) in complex with the CegC3 peptide (magenta). See Methods section for modeling. **b** Zoom in into CegC3 (magenta) peptide interactions with DotM (gray). Interacting residues are shown in stick representation, color-coded blue and red for N and O atoms, and either magenta or gray for C atoms of the peptide and the protein, respectively. Panels 1–3 focus on the three areas, M1, M2, and M4, respectively, which were targeted for mutations, and for which binding was abrogated

infection, other bacteria that harbor a DotM-containing T4BS system could also be targeted by such a compound. As DotM is a unique member of the T4BS systems class, yet highly conserved, the design of an antibiotic, which can be exploited against a broader spectrum of bacteria, might be possible.

## Methods

**Strains and constructs.** *Legionella* and *E. coli* strains used in this study are listed in Supplementary Table 1[6,25]. *Legionella* were grown on charcoal yeast extract (CYE) plates containing appropriate antibiotics (100 µg ml⁻¹ streptomycin and 10 µg ml⁻¹ chloramphenicol) as described previously[25]. *E. coli* were grown using standard media as described below.

Cloning of DotM fragments and mutants was carried out using the oligonucleotides listed in Supplementary Table 1. Full-length DotM was cloned from the genome of the *L. pneumophila* Philadelphia strain Lp01[6] into a pASK-IBA backbone vector (Chloramphenicol resistance) with sequence encoding a hexahistidine tag at the 5′ end in order to generate a His-tagged fusion. The construct was cloned by a PCR reaction (Phusion flash high-fidelity polymerase, Thermo Scientific) and inserted by the In-Fusion reaction (Clontech).

Initial analysis of DotM sequence using a transmembrane topology predictor (MEMSAT) revealed a membrane domain containing three membrane-spanning helices consisting of residues 19–37, 43–58, and 90–113. The soluble cytoplasmic domain of the protein (residues 119–380), denoted DotM119, was cloned. DotM119 construct was amplified by PCR and inserted into a pET backbone vector (kanamycin resistance) with a sequence encoding a hexahistidine tag at 5′ end, using In-Fusion (Clontech). Further on, expression and purification of DotM119 revealed a truncated form (starting at residue 153 according to EDMAN degradation), hence a new construct was cloned, denoted DotM153. In this construct, a Human Rhinovirus (HRV) 3C Protease cleavage site was introduced between the hexahistidine tag and DotM153.

Site-directed mutagenesis of surface arginines to glutamates (R196E/R197E, R314E/R315E, R347E/R348E, R217E, and R262E) was carried out on DotM153 gene using the Phusion flash high-fidelity polymerase (Thermo Scientific). The procedure included PCR using specific complementary primers (Supplementary Table 1) followed by a DpnI digestion.

For introduction of *dotM* mutants into *Legionella* using allelic exchange, *dotM* was first cloned in the pSR47S vector[37] including 1000 bp upstream and downstream to the *dotM* gene. Mutations were then introduced as described above. The resulting constructs were then used to introduce the *dotM* mutants into the *L. pneumophila* Lp01[6] genome by allelic exchange as previously described[38]. For the constructs of Cya fusion with effector signal peptides, the Cya domain sequence in the pMMB207 plasmid[25] was amplified using a reverse primer corresponding to the 3′ end of the Cya domain sequence, and a forward primer encoding for the same region of the 3′ end of the Cya domain sequence, followed by a short SGGGA linker and by the last 30 amino acids of various effectors (Supplementary Table 1). PCR products were re-circularized using In-Fusion. For the construct of the Cya fusion with full-length LegC8, the full-length *legC8* gene was amplified from the *L. pneumophila* genome, and introduced so as to produce a C-terminal fusion with a SGGGA linker as described above.

**Expression and purification of DotM proteins.** DotM119 and DotM153 were transformed into *E. coli* BL21 BLR(DE3) (Novagen) competent cells, and cells were cultured in 2XYT auto-induction media[39]. When OD ($\lambda = 600$ nm) reached ~0.7, protein production was induced with 0.4 mM isopropyl 1-thio-$\beta$-D-galactopyranoside (IPTG) (although we used an auto-induction media, increased expression was nevertheless observed by the addition of IPTG), and cells were incubated overnight at 18 °C. Cells were harvested and resuspended in lysis buffer (0.5 M NaCl, 40 mM Tris pH 8, 4% glycerol, 1 mM EDTA) with addition of lysozyme and DNase I (NEB), and homogenized by high pressure (40,000 psi) homogenization. The lysate was centrifuged at 61,000 × *g* for 30 min to remove the cell debris.

For DotM119, supernatants were loaded onto a HisTrap HP 5 ml (GE Healthcare) affinity column. The column was washed first with buffer A (0.5 M NaCl, 40 mM Tris pH 8, 4% glycerol, 1 mM EDTA) and then with buffer A containing 50 mM imidazole (Acros Organics) before applying a linear gradient up to 500 mM of imidazole to elute DotM proteins. Fractions containing DotM proteins were pooled and loaded on a Superdex200 16/60 size-exclusion chromatography column equilibrated in buffer A.

DotM153 was also purified via a HisTrap HP 5 ml affinity column as described for DotM119. However, after the first purification step, HRV-3C Protease (Pierce) was added to remove the His-tag by incubation overnight, while the protein was dialyzed against buffer A containing 1 mM DTT. The cleaved protein was reloaded on the HisTrap column. The unbound fraction was taken for further purification by applying it to a Superdex75 16/60 size-exclusion chromatography column (GE Healthcare) using a buffer consisting of 0.5 M NaCl, 50 mM Tris pH 7.6, 4% glycerol, and 1 mM EDTA. DotM153 mutants were expressed and purified as the wild-type protein. All purifications were performed at 4 °C.

Expression of SeMet-substituted protein was achieved by growing *E. coli* BL21 BLR(DE3) (Novagen) in 2XYT media until optical density at 600 nm reached ~1. Cells were washed twice in minimal media before suspension in half the initial

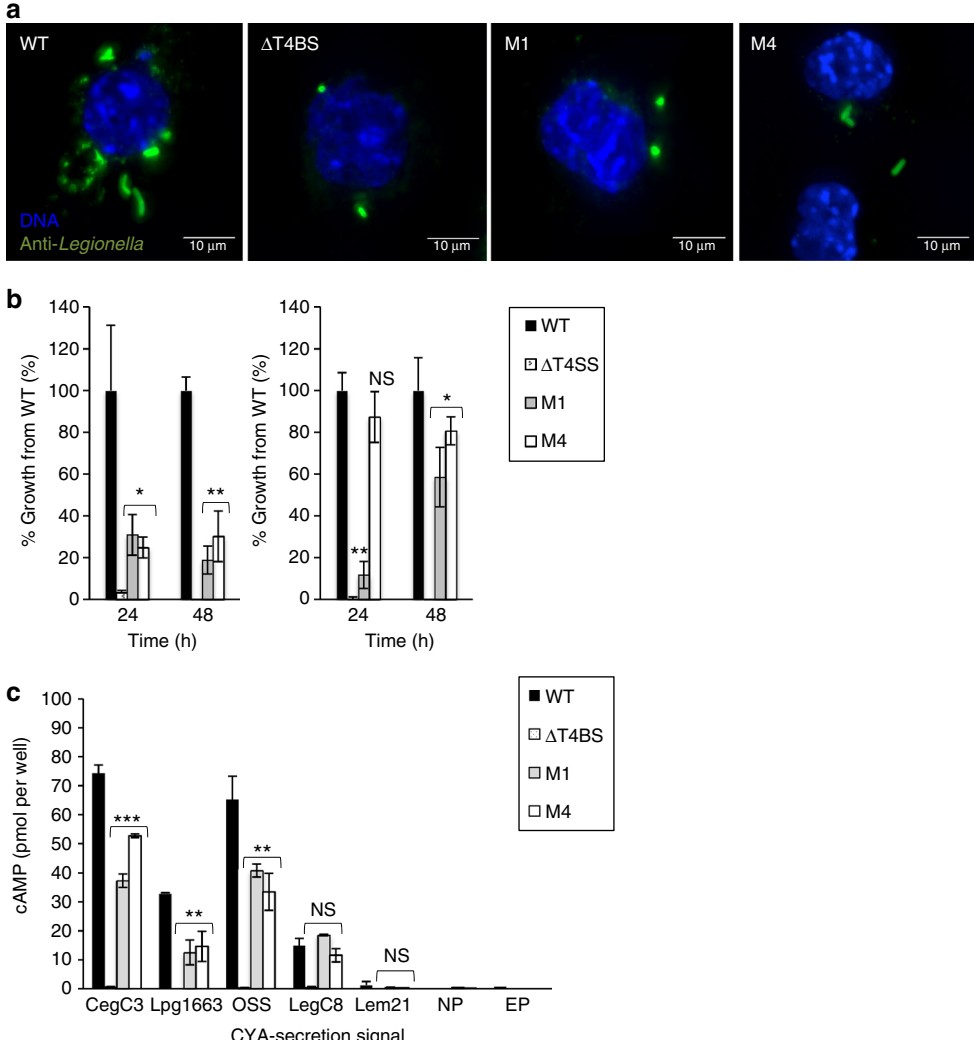

**Fig. 6** Survival and transfer activity of binding-defective DotM *Legionella* mutants. **a** J774A.1 mouse macrophage-like cells were infected for 2 h with wild-type (WT) *L. pneumophila* (Lp01), T4BS system-deficient *L pneumophila* (ΔT4BS) or isogenic strains containing the indicated *dotM* mutations (M1 or M4), and fixed using 4% PFA. Bacteria were stained with anti-*L. pneumophila* antibody and a secondary antibody labeled with Alexa fluor 488 (green), while the host cell DNA was stained using DAPI (blue). Stacked images are shown in each panel. Bar = 10 μm. **b** Intracellular growth of *L. pneumophila* strain Lp01 and isogenic strains producing the binding-defective DotM proteins M1 and M4. The intracellular growth of these *L. pneumophila* strains were determined in J774A.1 cells (left panel), and the protozoan host *A. castellanii* (right panel) as described in Methods. Graphs report colony-forming units ± standard deviation for each strain as percentages compared to wild-type. Bacteria were counted after either 24 or 48 h post infection as described in Methods. *, **, and *** indicate *P* values <0.05, <0.01, and <0.001, respectively. **c** Translocation of Cya fusions by *L. pneumophila* into host eukaryotic cells. The ability of various effectors C-terminal translocation signals to translocate a reporter Cya was monitored in DotM wild-type or binding-defective mutants *Legionella* backgrounds by measuring cAMP levels in host cells 90 min after infection as described in Methods. Translocation levels are shown for the *L. pneumophila* wild-type (WT) strain, the T4BS system-defective strain (ΔT4BS), and the *L. pneumophila dotM* mutants M1 and M4, as well as for bacteria without a plasmid (NP) and with unfused Cya (EP; shown only for WT). *, **, and *** indicate *P* values <0.05, <0.01, and <0.001, respectively

volume of the 2XYT media and grown at 37 °C for 30 min without methionine. Seleno-methionine was added and growth at 37 °C was continued for a further 30 min. The addition of 0.5 mM IPTG was followed by growth at 23 °C overnight. Purification of the SeMet DotM was carried out using the same procedure as previously detailed for native DotM153.

**Crystallization of DotM153 and its mutants**. Wild-type DotM153, its mutants M1, M2, and M4 and the SeMet derivative (all 15 mg ml⁻¹) crystals were obtained using the sitting drop method. The reservoir conditions yielding the best diffracting conditions were as follows: wild-type DotM153 crystallized in 0.1 M sodium citrate pH 5.0 and 15% w/v PEG 6000; the SeMet derivative DotM153 crystallized in 0.1 M MES pH 6.0, 0.15 M NH₄SO₄ and 25% w/v PEG 4000; DotM153 M1 crystallized in 0.2 M lithium sulfate, 0.1 M MES pH 6.0 and 20% w/v PEG 4000; DotM153 M2 crystallized in 0.1 M potassium chloride, 0.1 M HEPES pH 7.5 and 15% w/v PEG 6000; DotM153 M4 crystallized in 0.2 M potassium iodide, 0.1 M MES pH 6.5 and 25% w/v PEG 4000.

**Structure determination of DotM153 and DotM153 variants**. Crystals were flash-cooled in cryoprotectant solutions containing the original crystallization solution plus 25% (v/v) glycerol. Crystals of native DotM153 diffracted to 1.85 Å in the hexagonal space group P6₅ with the cell parameters $a = 118.5$ Å, $b = 118.5$ Å, $c = 66.3$ Å, $\alpha = 90.00°$, $\beta = 90.00°$, $\gamma = 120.00°$ (Table 1). Crystals of SeMet DotM153 diffracted to 2.15 Å in the same space group (P6₅) and cell parameters were $a = 118.9$ Å, $b = 118.9$ Å, $c = 66.7$ Å, $\alpha = 90.00°$, $\beta = 90.00°$, $\gamma = 120.00°$. For phasing, a single-wavelength anomalous dispersion (SAD) data set at the Se peak wavelength ($\lambda = 0.979$ Å) was collected at the Soleil Synchrotron (Soleil, L'Orme des Merisiers Saint-Aubin, France) at beamline PROXIMA-1.

For the SAD data set, data were processed using the XDS suite[40], and merged and scaled with Aimless[41]. SHELXC/D/E suite[42,43] was used to solve DotM153₍SeMet₎ structure. The search for anomalous peaks using SHELXD resulted in 22 strong peaks (occupancies larger than 30%), indicating the presence of two DotM153 monomers in the AU (the sequence of DotM153 contains 11 methionines). The correct space group (P6₅) could be distinguished by comparing the connectivity

and contrast value, after the first round of density modification and refinement of the heavy atom sites using SHELXE. The initial SAD map at the resolution range of 50–2.15 Å was traced by Buccaneer[44], which built ~90% of the two DotM molecules (408 residues), in addition to two shorter chains (19 amino acids each), which were later fitted into the missing N-terminal tail of monomers A and B. Further model building was carried out using Coot[45]. The SeMet structure was refined to 2.15 Å resolution using REFMAC5[46] implemented in the CCP4i suite[47]. For the structure of DotM153 mutants, molecular replacement was conducted using the refined DotM153 structure in Molrep[48]. These structures were further built and refined by Coot and REFMAC5, respectively.

**DotM–CegC3 peptide complex modeling.** The apo DotM monomer structure, along with the sequence of CegC3 peptide, was input to the CABS server[32]. Out of the 10 best models, those docking CegC3 in agreement with the biochemistry results were elected for further work. The elected structure was taken for further refinement in FlexPepDock server[33,34].

**Isothermal titration calorimetry studies.** Isothermal titration calorimetry (ITC) experiments were conducted at 25 °C using MicroCal ITC200. All proteins and peptides were dialyzed into ITC buffer (100 mM NaCl, 25 mM HEPES pH 7.5), and experiments were performed with final protein and peptide concentrations of 10–30 and 220–350 μM, respectively. The peptide solution (in the syringe) was titrated into the protein solution (in the sample cell). Titrations consisted of 19 consecutive 2 μL injections (following a pre-injection of 0.4 μL) into the protein sample at 150 s intervals. The initial injection was discarded from the data analysis. Control experiments consisted of injecting peptides into buffer and showed no significant heat of dilution. The data were processed and thermodynamic parameters obtained by fitting the data to a single-site-binding model using the Origin software. DotM153 mutants were treated identically to DotM wild-type and analyzed for their interactions with CegC3, Lpg1663, and OSM peptides only.

**Cell culture.** CHO FcγRII cells (as previously described[49]) and J774.1A (ATCC TIB-67) macrophage-like cells were cultured at 37 °C in 5% $CO_2$ in α-MEM plus 10% FBS and RPMI-1640 plus 10% FBS, respectively. *A. castellanii* (ATCC 30234) were cultured routinely at room temperature in ATCC medium 712 (PYG).

**Legionella intracellular growth in eukaryotic hosts.** Growth in protozoan host was monitored as described[50]. For intracellular growth in macrophages, the J774A.1 cell line was used. Cells were plated at $1 \times 10^5$ cells per well onto 24-well tissue culture dishes. The cultures were infected with $1 \times 10^5$ bacteria (multiplicity of infection (MOI) = 1), followed by centrifugation for 5 min at $200 \times g$ at room temperature and incubation at 37 °C in a 5% $CO_2$ atmosphere for 2 h. Excess bacteria were then washed away using PBS and plate-adhering infected cells were incubated in fresh medium for a further 24 or 48 h at 37 °C in a 5% $CO_2$ atmosphere. For CFU determination, the medium in each well was collected, cells were lysed in sterile water, and resulting cell lysates were combined with the removed cell culture medium. The resulting mixtures were diluted in water, plated on CYE agar plates, and incubated at 37 °C for CFU determination.

**Fluorescence microscopy of infected cells.** Sterile glass coverslips were placed in each well of a 24-well tissue culture plate, and $7.5 \times 10^4$ J774A.1 cells were added to each well in DMEM medium 24 h prior to infection. Cells were then infected with *L. pneumophila* from a 48 h heavy patch with an estimated MOI of 20 and infections were synchronized by spinning plates at $200 \times g$ for 5 min, followed by an incubation at 37 °C. 1, 4, and 8 h after infection cells were fixed with 4% PFA for 10 min. *L. pneumophila* cells were stained using a rabbit anti-*Legionella* antibody[49] (1:1000) followed by an Alexa Fluor 488-conjugated secondary antibody (Invitrogen, 1:2000) and then washed extensively with PBS. 4,6-diamidino-2-phenylindole staining (DAPI, 0.1 μg ml$^{-1}$, Life Technologies) was used to identify host cell nuclei and bacterial cells. In all experiments coverslips were mounted using the ProLong gold antifade reagent (Invitrogen). Coverslips were imaged on a Nikon Eclipse TE2000-S inverted fluorescence microscope with a ×100/1.4 numerical aperture objective lens. The microscope camera was a Photometrics CoolSNAP EZ camera controlled by SlideBook™ (Intelligent Imaging Innovations).

**Cya assay.** CHO FcγRII cells ($1 \times 10^5$ cells per well) were placed into 24-well tissue culture plates 1 day prior to infection. The cell culture medium was aspirated before adding to each well the *Legionella* cells ($1.5 \times 10^6$ bacteria per well) engineered to contain the DotM mutants and the Cya-containing fusions. Rabbit anti-*Legionella* antiserum diluted at a ratio of 1:1000 (which facilitates *Legionella* adhesion) and 0.5 mM IPTG (to induce Cya fusions) were also added. The plates were centrifuged onto a confluent monolayer of host cells for 5 min at $200 \times g$, warmed in a 37 °C water bath for 5 min, then placed in a $CO_2$ incubator for 1.5 h. Cells were washed three times with ice-cold PBS and lysed in 200 μl of a solution containing 50 mN HCl and 0.1% Triton X-100 on ice. After boiling for 5 min, 12 μl of 0.5 M NaOH were added and cAMP was extracted with 2 volumes of ethanol. Insoluble materials were pelleted by centrifugation, and the cAMP-containing soluble material was lyophilized. The cAMP levels were determined for each extract

by using an ELISA kit according to manufacturer's instructions (Amersham Biosciences, RPN-225).

**Statistical analysis.** Statistical analysis was performed with GraphPad Prism v.5.0 (GraphPad Software, La Jolla, CA, USA). For comparison of two groups, an unpaired *t*-test was employed. A *P* value of <0.05 was considered statistically significant. All experiments were performed at least three times. The data are expressed as mean ± standard deviation.

**Data availability.** The atomic coordinates of DotM153 SeMet derivative, wild-type and mutants M1, M2, and M4 structures have been deposited in the Protein Data Bank with accession codes 6EXD, 6EXB, 6EXA, 6EXC, and 6EXE, respectively. All remaining data can be obtained from the corresponding author upon reasonable request.

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

## Acknowledgements

This work was funded by ERC grant 321630 to G.W. We thank Professor Joseph Vogel for kindly providing DotM antibodies. We thank the staff of Diamond light Source (beamlines I02 and I24), SOLEIL (beamline PROXIMA-1), and ESRF (beamline ID23-1) for data collection, Dr Ambrose Cole (Birkbeck College) for his help with data collection and analysis, and Talha Arooz (UCL) for assistance with the ITC measurements.

## Author contributions

A.M. and G.W. conceived the study and initiated the project. G.W. and C.R. supervised the project. A.M. performed and analyzed crystallography experiments including protein expression and purification of DotM and its variants, crystallization, structure determination and analysis, isothermal titration calorimetry, and structural modeling. A.M. also generated *Legionella dotM* mutant strains, performed *Legionella* infections, immunoprecipitation, and Cya translocation experiments and analyzed the data. D.C. and L.L. performed Cya translocation experiments and analyzed infection data. D.C. also performed immunoprecipitation experiment. A.M., G.W., and C.R. wrote and all authors edited the manuscript.
