## [Peer Review File · Nature Communications]

Reviewers' comments:

Reviewer #1 (Remarks to the Author):

DotM structure reveals a role in effector recruiting by the Type 4B secretion system of *Legionella pneumophila*

Amit Meir, David Chetrit, Craig Roy, Gabriel Waksman

In this study, the authors solve the structure of the cytoplasmic domain of DotM, defining both a novel protein fold and, importantly a series of surface-exposed positively charged regions that may function as a docking site for the negatively charged glutamate rich signal sequence of a subset of Dot/Icm translocated effectors. They present evidence of binding between DotM and effector signal-containing peptides that correlates with the presence of a glutamate rich motif and test the importance of the DotM surface-exposed arginine residues for binding through mutagenesis studies. Finally, the authors examine the efficiency of effector translocation and *Legionella* replication during infection with strains expressing allelic variants of DotM.

The work is significant in that it is the first structure of a functional domain of DotM and attempts to address a key question in the field, how are effectors selected and/or targeted for translocation. It is a very interesting idea with significant merit and potential impact. However, while some of the data is consistent with the authors' conclusion that "DotM forms the interacting surface for recruitment of Glu-rich motif-containing *Legionella* effectors", the data is not conclusive.

Major comments:

1. The conclusion that DotM binds Glu-rich containing effectors is based on DotM binding a single Glu-rich motif-containing peptide and an artificial peptide sequence (OSM). One biologically relevant example is a gross underrepresentation of the collection of effectors belonging to this class. A more comprehensive experimental dissection of this interaction (see Comment 3) or additional examples would be more convincing.
2. The authors test two IcmSW-independent effectors, one that has a Glu-rich motif (LegC 3) and one that consists of several equally spaced glutamate residues (Lem8) (Table 2) that they claim is not a Glu-rich motif. The author's definition of a Glu-rich motif is unclear. As an extension of this, the glutamate rich signal driving effector translocation was originally defined and termed an E-block by Isberg and colleagues (as referenced by the authors), is there a reason the authors have opted to rename this region? Are the two classes, Glu-rich motifs and E-blocks different?
3. How do the authors conclude that lack of binding of LegC 3 short to DotM is due to a requirement for the N-terminal glutamates and not stability or solubility of the peptide? The authors state that they included two glycine residues at the N-termini of the full length peptides (Table 2) to increase solubility but these are absent from LegC 3 short. Was the solubility/stability of LegC 3 tested? Docking studies of the LegC 3 signal peptide to DotM and the lack of binding of LegC 3 short predict that LegC 3 glutamate at position 145 plays a central role in binding DotM. Demonstrating that this is the case through binding studies between a LegC 3 E145A variant and DotM, and wild type LegC 3 with DotM R196A (or R196E) would help solidify the author's conclusion and circumvent any discrepancies in solubility of LegC 3 short.
4. How do the authors separate defects in Dot/Icm complex assembly and stability from recognition and translocation of Glu-rich motif-containing effectors (function)? The similarities in structure

between DotM and the M1 and M4 variants are limited to a portion of the full length protein. While allelic replacement of dotM with the M1 and M4 alleles are consistent with the claim that DotM variants do not destabilize the translocon, the detrimental effect of amino acid substitutions at R314 and R315 in the same vicinity and the defect in translocation of LegC 8 and Lem21 (lacking Glu-rich motifs) by the M1 and M4 variants, similar to LegC 3 and Lpg1663 (having Glu-rich motifs) (Fig. 6C) suggest otherwise. The authors gloss over this latter result suggesting the effect is minor given the low basal level of translocation in wild type cells. Yet, the reported statistically significant difference contradicts their conclusion and needs to be addressed. Demonstrating that the M1 and M4 variants do not affect translocon assembly/stability (DotL and DotN are stably expressed at wild type levels) or function (translocation of effectors lacking Glu-rich motifs at levels comparable to bacteria harboring wild type DotM) would be more convincing of the model. For example, is there a significant difference in LegC 8 and Lem21 dependency on M1/M4 and IcmSW for translocation when full length LegC 8 and Lem21 are fused to Cya? Are these fusions translocated with the same efficiency independent of whether wild type DotM or the M1 or M4 variants are expressed?

5. While the consensus signal defined by Liftshitz et al., is a run of 8 glutamate residues, there are few, if any effectors with this sequence. Instead, the majority seem to consist of regularly spaced glutamates or glutamate pairs (EExxE or EExEENxNS as defined by Huang et al., 2011) and, in the case of SidM (Zhu et al., 2010), this appears to create a negatively charged surface on one side of an alpha helix. While modeling studies of the LegC 3 peptide docking with DotM supports their model, it's based on a predicted structure of LegC 3. A similar analysis with an effector for which the structure has been solved, such as SidM, would substantiate the conclusions. Does the 3-D organization of the glutamates, especially those shown to be important for translocation (Huang et al., 2012), facilitate docking of the SidM signal to the binding surface of DotM?

6. The authors state "remarkably, given that Glu-rich motif-containing effectors represent only a subset of Legionella effectors, the M1 and M4 mutations resulted in a significant decrease in *L. pneumophila* replication". Could the authors be more definitive here? What percentage of effectors does this encompass? Is this surprising, especially since one of the Glu-rich motif proteins is IroT/MavN, which is essential for Legionella replication in macrophages and amoebae (Portier, 2015; Isaac, 2015)?

7. In the introduction and the results, the authors elude to a division between effectors in which there are IcmSW-independent effectors that have glutamate rich motifs and thus are likely to be DotM-dependent, and IcmSW-dependent effectors that lack glutamate-rich repeats that are thus likely DotM-independent. Are the glutamate rich motifs and IcmSW-dependent signals truly mutually exclusive? Given that a very large number of effectors have not been tested for IcmSW dependence, some clarification here would be helpful.

8. In Figure 6B, the authors indicate a statistical difference between Legionella harboring wild type DotM and the DotM M4 variant for replication in *A. castellanii* at 24 and 48 hours. Based on the percent growth and error bars this seems unlikely. Can the authors please verify the statistical analysis here is correct?

9. The discussion focuses largely on speculating about what other Dot/Icm components may be involved in effector translocation based on charge. A more detailed discussion of the data and its integration into the proposed model would be more informative.

10. The differential importance of M1 and M4 putative docking sites between the two host cell types is interesting: do the authors have an explanation for this?

Minor Comments:

1. As written, it is unclear where the OSM sequence comes from – a single sentence providing details of its origin would be very helpful.
2. Proteins are not mutated. Please revise the text throughout to amino acid substitutions, variants or some appropriate equivalent.
3. The term effector is used loosely. Effector is restricted to translocated proteins for which an effect on the host cell has been defined. For Legionella, this is only the case for 10-15% of translocated proteins.
4. Please provide a reference demonstrating that “The subsequent death of the macrophage signals the beginning of Legionnaire’s disease, ...” or remove.
5. There are several abbreviations in the text that have not been defined. Please revise.
6. It is unclear what Fig. 1C provides to the manuscript and could be removed.
7. Remove quotation marks for all terms.
8. Line 76: “trans(inner)membrane” should read “integral inner membrane protein”.
9. Line 80: “the DotL’s C-terminus” should read “the C-terminus of DotL”
10. Line 41: Legionella should be italicized.
11. Line 124-125: “both N and C terminal” should read “both N- and C-terminal”.
12. Line 144: “While structure homology search” should read “While the structure homology search”.
13. Line 15-155: “27-30 amino-acids long, derived from the C-terminus of” should read “27-30 amino acids long, derived from the C-termini of”.
14. Line 178-179: remove “see location of the targeted residues in”

Reviewer #2 (Remarks to the Author):

The manuscript by Meir et al presents very interesting and novel information regarding the way by which the Legionella pneumophila Dot/Icm complex recognizes a subset of its effectors using the DotM protein. The authors present the crystal structure of DotM which contains a large patch of positively charged amino acids residues. This large patch of positively charged residues was found to be important for the binding of the Glu-rich signal motif located at C-terminus of some effectors. This finding was validated using site directed mutagenesis of the DotM positively charged residues and by binding of the OSM synthetic signal as well as effectors C-terminus to DotM. In addition, the DotM mutants were introduced into the L. pneumophila genome and examined for their effect on intracellular growth and effector translocation using the CyaA translocation system. The DotM mutations were found to severely affect intracellular growth as well as the translocation of C-terminal 30 residue sequence of two L. pneumophila effectors. The manuscript is very well written and easy to

follow and the in vitro experiments are well performed. However, the in vivo work requires few additions and controls in order to strengthen the conclusions made.

Major comments

1. The translocation analysis using the DotM mutants M1 and M4 – the authors should examine if the DotM mutants are stable in-vivo. As indicated in lines 188-190, the authors crystalize the mutants and their structures were found to be virtually identical to the wild-type protein. However, the mutations might still affect the stability of the proteins in-vivo, or the ability to bind the other two components of the coupling complex DotL or DotN. A simple western using DotM specific antibody can address this point, or generating a genomic tagged version of DotM in case antibodies are not available.

2. The translocation analysis of "IcmSW-dependent effectors" using wild-type DotM and the M1 and M4 mutants – the use of only the C-terminal 30 residues sequence of the effectors (Lem21 and LegC8) in this analysis, is problematic. As the authors indicated, the weak translocation of these effectors might result from the absence of internal IcmSW binding sites. Therefore, this analysis is not informative. However, it is very important to determine whether the DotM mutants affect the translocation of IcmSW-dependent effectors or not. The authors should examine full length constructs of these two effectors using the CyaA system in order to resolve this issue, since this point is important for the understanding of effector translocation by the Dot/Icm complex.

Minor comments

1. Fig. 6B – for clarity, please divide panel 6B into two separate panels one for J774 and one for *A. castellanii*.

2. Conclusions, line 304, it is reasonable to assume that most of the dot/icm mutants will be found to be defective for translocation and not just the icmT mutant which was examined. Please rewrite the sentence, referring more to the positively charged residues of IcmT and less to its translocation phenotype.

Reviewer #3 (Remarks to the Author):

DotM is a component of the T4BSS present in various pathogenic intracellular bacteria. In contrast to T4ASS, structural information on T4BSSs is limited. In this manuscript the authors present the crystal structure of DotM from *Legionella pneumophila* to a resolution of 1.8 Å, which together with DotL and DotN forms the so-called coupling complex in T4BSSs. The authors use a traditional X-ray crystallography approach for structure determination. The structure of DotM reveals charged surface regions that might facilitate binding of effector proteins to be exported via the membrane-spanning components of secretion system. To confirm this hypothesis the authors analyzed the interaction of DotM with peptides containing Glu-rich secretion motifs in vitro and the effect of DotM variants with disrupted charged surface regions in vivo.

Given the importance for virulence of a number of human pathogens, any structural information on T4BSSs is of interest for a broader audience. Although the structure of DotM in complex with other T4BSS components would greatly enhance this study in general, I understand that complex preparation, crystallization and structure determination would most likely take months to years, and is therefore outside of the scope of this manuscript.

The experiments presented in this study appear to have been carefully done. The manuscript is well written and conclusions should be understandable to non-specialists. The introduction section covers current literature on the topic appropriately. The method section is very clear and includes enough details to enable reproduction of the experiments. However, I feel that several points and questions need to be addressed to before this manuscript can be accepted for publication in Nature

C ommunications.

1. Is the observed interaction between DotM and the peptides containing secretion motifs sequence specific? Do peptides of identical length but with a random sequence of negatively charged residues bind to DotM, too? The authors should perform appropriate ITC experiments to investigate if the interaction of effectors with DotM is solely based on charge or if the sequence motif found in Legionella effector proteins is necessary .

2. The authors note that attempts at introducing M2 mutations might have failed due to a destabilization of the DotMLN coupling complex , however, the Δ T4BS strain is viable. This should be further explained.

3. Based on the presented results it cannot be excluded that effector translocation defects of DotM variants with disrupted charged surface patches are a result of a defective formation of the T4BSS complex itself. This needs to be excluded to strengthen the authors' hypothesis that the observed effect is indeed based on defective effector binding.

4. The presented model for the C egC3-interaction with DotM153 was manually chosen to match the obtained ITC data. It does not represent an independent result and therefore cannot be used to strengthen the proposed interaction between Glu-rich peptides and DotM. Although this is briefly mentioned in the materials and methods section my feeling is that it should be included in the main text as well.

5. The authors state that in solution DotM153 and DotM119 forms monomers and dimers, respectively. The asymmetric unit of DotM119 crystals contains 2 molecules. Do those molecules form a stable dimer that was observed in solution? The authors should clarify (or at least comment) on the physiological relevance of the different oligomeric states of DotM.

6. I annotated a small number of typos in the attached pdf document.

[Editorial Note: Annotated PDF Document not included in Peer Review File due to journal embargo policy]

-

Dr. Guido Hansen
Institute of Biochemistry
University Lübeck
Tel +49 451 3101 3122
E-Mail hansen@biochem.uni-luebeck.de

Reviewer #1

In this study, the authors solve the structure of the cytoplasmic domain of DotM, defining both a novel protein fold and, importantly a series of surface-exposed positively charged regions that may function as a docking site for the negatively charge glutamate rich signal sequence of a subset of Dot/Icm translocated effectors. They present evidence of binding between DotM and effector signal-containing peptides that correlates with the presence of a glutamate rich motif and test the importance of the

DotM surface-exposed arginine residues for binding through mutagenesis studies. Finally, the authors examine the efficiency of effector translocation and Legionella replication during infection with strains expressing allelic variants of DotM.

The work is significant in that it is the first structure of a functional domain of DotM and attempts to address a key question in the field, how are effectors selected and/or targeted for translocation. It is a very interesting idea with significant merit and potential impact. However, while some of the data is consistent with the authors' conclusion that "DotM forms the interacting surface for recruitment of Glu-rich motif-containing Legionella effectors", the data is not conclusive.

Major comments:

Comment 1:

The conclusion that DotM binds Glu-rich containing effectors is based on DotM binding a single Glu-rich motif-containing peptide and an artificial peptide sequence (OSM). One biologically relevant example is a gross underrepresentation of the collection of effectors belonging to this class. A more comprehensive experimental dissection of this interaction (see Comment 3) or additional examples would be more convincing.

Response:

To respond to this excellent suggestion, we carried out additional ITC experiments on one additional Glu-rich motif-containing peptide, that of the effector Lpg1663 (a highly ranked signal peptide in Lifshitz et al scoring). The ITC experiment is now reported in Figure 4. As can be seen, DotM binds the Lpg1663 motif with high affinity. We also carried out ITC experiments with the M1 and M2 mutants of DotM, for which we have demonstrated that they are no longer able to bind the Glu-rich motif of CegC3. We show here (reported in Figure 4) that these mutants are also defective in binding the Glu-rich motif of Lpg1663, adding another demonstration that DotM binds Glu-rich sequences of *Legionella* effectors. We feel that adding yet another example would be pointless since, with Lpg1663, we confirm the results obtained using the CegC3 and OSM peptides.

Comment 2:

The authors test two lcmSW-independent effectors, one that has a Glu-rich motif (LegC3) and one that consists of several equally spaced glutamate residues (Lem8) (Table 2) that they claim is not a Glu-rich motif. The author's definition of a Glu-rich motif is unclear. As an extension of this, the glutamate rich signal driving effector translocation was originally defined and termed an E-block by Isberg and colleagues (as referenced by the authors), is there a reason the authors have opted to rename this region? Are the two classes, Glu-rich motifs and E-blocks different?

Response:

We thank the reviewer for pointing this out. What DotM-binding appears to require is **acidic** signal sequences. Indeed, all peptides that we have studied and that are positive for DotM binding have a pI of 4 or below. Lem8 contains many Glu residues in its signal peptide and can be considered Glu-rich but its pI is 6.3, i.e. close to neutral. So what appears to be the defining feature of DotM-binding is Glu-rich sequences that have very low pI. Therefore, we have renamed the motifs mediating binding to DotM as **acidic Glu-rich motifs**. We have now corrected the text throughout and defined "acidic Glu-rich motifs" on page 8 by saying: "*Lem8, which received a high score according to statistical calculation by Lifshitz et al., did not bind either; its sequence contains a large number of Glu residues but its overall pI is 6.3 because acidic residues are neutralized by an equal number of adjacent Lys. Thus, DotM is able to bind Glu-rich peptides with high affinity, provided that the overall pI of these peptides is low: we termed this subset of Glu-rich peptides "acidic Glu-rich motifs/peptides."*

Comment 3:

How do the authors conclude that lack of binding of LegC3 short to DotM is due to a requirement for the N-terminal glutamates and not stability or solubility of the peptide? The authors state that they included two glycine residues at the N-termini of the full length peptides (Table 2) to increase solubility but these are absent from LegC3 short. Was the solubility/stability of LegC3 tested?

Docking studies of the LegC3 signal peptide to DotM and the lack of binding of LegC3 short predict that LegC3 glutamate at position 145 plays a central role in binding DotM. Demonstrating that this is the case through binding studies between a LegC3 E145A variant and DotM, and wild type LegC3 with DotM R196A (or R196E) would help solidify the author's conclusion and circumvent any discrepancies in solubility of LegC3 short.

Response:

We have now revisited these experiments with a CegC3 short peptide with two glycine residues at the N-terminus. We then carried out ITC experiments with this peptide and shows that it is not binding (see new Figure 4). Thus solubility is not an issue. Furthermore, we have carried out ITC binding experiments using a CegC3 peptide mutated at residue 145 to Ala (CegC3 E145A) and we show that this peptide no longer binds DotM, confirming that the interaction in which residue CegC3 E145 is involved is indeed crucial. This experiment validates our structural model. We have added on pages 9 and 10: *“Other interactions are between residues R196 and R197 of DotM and CegC3 E145 and a stacking interaction between these two residues and the side chain of CegC3 F146 (Figure 5). To validate this model, E145 was mutated to Ala and, using ITC, this mutant CegC3 peptide was shown to no longer bind to DotM (Figure 4).”*

Comment 4:

How do the authors separate defects in Dot/Icm complex assembly and stability from recognition and translocation of Glu-rich motif-containing effectors (function)? The similarities in structure between DotM and the M1 and M4 variants are limited to a portion of the full length protein. While allelic replacement of dotM with the M1 and M4 alleles are consistent with the claim that DotM variants do not destabilize the translocon, the detrimental effect of amino acid substitutions at R314 and R315 in the same vicinity and the defect in translocation of LegC8 and Lem21 (lacking Glu-rich motifs) by the M1 and M4 variants, similar to LegC3 and Lpg1663 (having Glu-rich motifs) (Fig. 6C) suggest otherwise. The authors gloss over this latter result suggesting the effect is minor given the low basal level of translocation in wild type cells. Yet, the reported statistically significant difference contradicts their conclusion and needs to be addressed. Demonstrating that the M1 and M4 variants do not affect translocon assembly/stability (DotL and DotN are stably expressed at wild type levels) or function (translocation of effectors lacking Glu-rich motifs at levels comparable to bacteria harboring wild type DotM) would be more convincing of the model. For example, is there a significant difference in LegC8 and Lem21 dependency on M1/M4 and IcmSW for translocation when full length LegC8 and Lem21 are fused to Cya? Are these fusions translocated with the same efficiency independent of whether wild type DotM or the M1 or M4 variants are expressed?

Response:

We thank the reviewer for his/her excellent suggestions. We first addressed the comment that M1 and M4 might destabilize the protein. Using DotM antibodies (kindly provided by Professor Joseph Vogel from Washington University in Saint Louis), we monitored the stability and expression level of the DotM and DotM variants in our strains. These results are now reported in Figure S2 and demonstrate that the mutated proteins are as stable as wild-type and also expressed in equal quantities as wild-type. We have added on page 10: *“Production of the DotM, M1 and M4 proteins in cells was*

monitored using anti-DotM antibodies and all wild-type and variant DotM proteins were shown to be produced in equal quantities (Figure S2A)."

Next, we revisited the statistical errors associated with the results of the experiments on the translocation of LegC8 and Lem21, and repeated the experiments, and found there is no statistically significant differences between wild-type, M1 and M4 for these effector signal sequences. So we amended the text accordingly and now say on pages 11 and 12: *"The Cya-LegC8_{Cter} and Cya-Lem21_{Cter} reporter fusions were translocated very weakly by all strains (Figure 6C) and there was not statistically-significant differences between strains"*.

Finally, to address the third issue raised by the reviewer, we produced a Cya-fusion of full-length effector LegC8 (lcmSW-dependent) and monitored translocation of this fusion protein in the DotM wild-type, M1 and M4 strains. These results are reported in Figure S2B, and show that i- translocation of the full-length Cya-LegC8 fusion is increased compared to Cya-LegC8_{Cter}, and ii- the mutations in DotM do not affect lcmSW-dependent transport of LegC8. Thus, our interpretation is correct. We have amended the text accordingly. It now reads on page 12: *"The Cya-LegC8_{Cter} and Cya-Lem21_{Cter} reporter fusions were translocated very weakly by all strains (Figure 6C) and there was not statistically-significant differences between strains. Weak translocation of these fusions is presumably due to the absence of internal lcmSW binding sites. Thus, to ensure that this is the case, full-length LegC8 was fused to the C-terminus of Cya (yielding a fusion termed Cya-LegC8) and translocation of Cya-LegC8 was monitored using the wild-type, M1 and M4 Legionella strains (Figure S2B). We observed higher levels of translocation, yet still no statistically-significant differences between wild-type and mutant strains, suggesting that weak translocation of Cya-LegC8_{Cter} is indeed due to the absence of an lcmSW-dependent translocation signal. Also, the fact that the full-length Cya-LegC8 fusion is translocated with the same efficiency in all Legionella strains indicates that the M1 and M4 mutations do not affect the lcmSW-dependent translocation function of the DotMLN complex."*

Comment 5:

While the consensus signal defined by Liftshitz et al., is a run of 8 glutamate residues, there are few, if any effectors with this sequence. Instead, the majority seem to consist of regularly spaced glutamates or glutamate pairs (EExxE or EExEENxNS as defined by Huang et al., 2011) and, in the case of SidM (Zhu et al., 2010), this appears to create a negatively charged surface on one side of an alpha helix. While modeling studies of the CegC3 peptide docking with DotM supports their model, it's based on a predicted structure of CegC3. A similar analysis with an effector for which the structure has been solved, such as SidM, would substantiate the conclusions. Does the 3-D organization of the glutamates, especially those shown to be important for translocation (Huang et al., 2012), facilitate docking of the SidM signal to the binding surface of DotM?

Response:

We note that, although there are glutamate pairs in the SidM's signal peptide sequence, the overall pI of the signal peptide (sequence QLLGLKTSSVSSFVKMVEETRESIKSQERQTIKIK) is 9.4 i.e. very basic. Also, SidM's Liftshitz score is 6.22 i.e. close to the arbitrary threshold of 5 for lcmSW-dependence. By contrast, CegC3's pI is 3.99 and its Liftshitz score is 16.55. Whether the SidM signal peptide binds DotM is not known and an experiment aiming at measuring the affinity between the two molecules is not requested by the reviewer; however, the reviewer requests that we model the known structure of the SidM signal peptide (which is part of a larger structure of SidM) onto our structure of DotM. We carried out this docking exercise using the programme ROSETTA, and obtained a model of the complex. This model is shown below. As can be seen, ROSETTA finds a docking site in DotM close to the CegC3 binding site, but makes very different interactions, being remote from M4 and M1, two crucial binding sub-sites for CegC3 binding. Our conclusion is that, if the signal peptide from SidM binds DotM, it does so in a different mode. More likely, given the high basic pI of the SidM signal

peptide, this peptide might not bind DotM. Given its low Lifshitz score, its transport might be lcmSW-dependent. Because there is no experimental data on DotM binding by the SidM sequence peptide and because the results of docking exercises are generally highly problematic unless they are validated (as we did with CegC3), we chose to not incorporate these data in the paper. As suggested by the reviewer in his/her comment 1, we have already added 1 peptide to our binding experiments: we don't think adding more will change our conclusions already amply substantiated by the study of 6 peptides and 5 effectors.

Docking of the SidM signal peptide onto the surface of DotM. DotM is in surface representation color-coded in grey while the SidM (left panel) and CegC3 (right panel) signal peptides are in semi-transparent surface representation color-coded in yellow and magenta, respectively. The positions of M1, M2, and M4 are shown. As can be seen, the SidM signal peptide docks on top of M2 but positions far from M1 and M4 and therefore is unable to make contacts with the corresponding residues.

Comment 6:

The authors state “remarkably, given that Glu-rich motif-containing effectors represent only a subset of Legionella effectors, the M1 and M4 mutations resulted in a significant decrease in L. pneumophila replication”. Could the authors be more definitive here? What percentage of effectors does this encompass? Is this surprising, especially since one of the Glu-rich motif proteins is IroT/MavN, which is essential for Legionella replication in macrophages and amoebae (Portier, 2015; Isaac, 2015)?

Response:

The signal peptide of IroT/MavN has a pI of 8.8 compared to a pI of 3.9, 4, or 3.6 for CegC3, Lpg1663 or OSM, respectively. Its score by Lifshitz et al is 5.31 against a score of 16.55 for CegC3. So we respectfully disagree with the reviewer's assessment. Concerning the reviewer's comment requesting us to be more specific regarding the percentage of effectors, the transport of which might be affected by mutations M1 or M4, it would be very difficult to assess rigorously. It would mean testing transport

of possibly 100s of effectors with acidic Glu-rich signal sequences, a task which is beyond the scope of this manuscript. We have however modified the conclusion of our manuscript to reflect this and now say: *“Our structural and biological investigations of DotM reveal an important role of DotM in recruitment of Legionella effectors of a particular class; our discovery opens new avenue of research aiming at determining exactly how many and which Legionella effectors might use DotM as a docking platform.”*

Comment 7:

In the introduction and the results, the authors elude to a division between effectors in which there are lcmSW-independent effectors that have glutamate rich motifs and thus are likely to be DotM-dependent, and lcmSW-dependent effectors that lack glutamate-rich repeats that are thus likely DotM-independent. Are the glutamate rich motifs and lcmSW-dependent signals truly mutually exclusive? Given that a very large number of effectors have not been tested for lcmSW dependence, some clarification here would be helpful.

Response:

We apologize for having conveyed such a dichotomy. The reviewer is absolutely right and we have added in the conclusion section: *“Although many effectors use either DotM or lcmSW as recruitment platforms, the use of these platforms might not be mutually exclusive and it is possible that some effectors might use both simultaneously.”*

Comment 8:

In Figure 6B, the authors indicate a statistical difference between Legionella harboring wild type DotM and the DotM M4 variant for replication in A. castellanii at 24 and 48 hours. Based on the percent growth and error bars this seems unlikely. Can the authors please verify the statistical analysis here is correct?

Response:

The reviewer is right: it is not statistically significant. We have corrected and now say: *“However, in A. castellanii, M1 reduces intracellular growth by up to 90% (after 24 hours) while the slight reduction in intracellular growth of the M4 mutant is not statistically significant.”*

Comment 9:

The discussion focuses largely on speculating about what other Dot/lcm components may be involved in effector translocation based on charge. A more detailed discussion of the data and its integration into the proposed model would be more informative.

Response:

We have indeed structured the paper differently, choosing to provide a conclusion, not a discussion. We wish to keep the same structure and therefore have not complied with this reviewer's recommendation.

Comment 10:

The differential importance of M1 and M4 putative docking sites between the two host cell types is interesting: do the authors have an explanation for this?

Response:

We now provide speculative explanations as to what could be happening. This is presumably due to different requirements for growth inhibition, some effectors signal sequences being more or less

dependent on the M4 site than others, and these differences being able to account for the differences in growth inhibition depending on the cell type. We now say: *“Intriguingly, M4 appears to inhibit growth much more markedly in macrophages, perhaps indicating differential effects of acidic Glu-rich motif-containing effectors depending on the host.”*

Minor Comments:

1. As written, it is unclear where the OSM sequence comes from – a single sentence providing details of its origin would be very helpful.

Response: we now have added a sentence.

2. Proteins are not mutated. Please revise the text throughout to amino acid substitutions, variants or some appropriate equivalent.

Response:

It is now been corrected.

3. The term effector is used loosely. Effector is restricted to translocated proteins for which an effect on the host cell has been defined. For Legionella, this is only the case for 10-15% of translocated proteins.

Response:

We believe we have done our best to correct this.

4. Please provide a reference demonstrating that “The subsequent death of the macrophage signals the beginning of Legionnaire’s disease, ...” or remove.

Response:

We have removed.

5. There are several abbreviations in the text that have not been defined. Please revise.

Response:

We have revised.

6. It is unclear what Fig. 1C provides to the manuscript and could be removed.

Response:

Fig 1C shows a region of the density with the model built in it. Most crystallographers rely on this kind of pictures to assess the structure. So we have left the figure as is.

7. Remove quotation marks for all terms.

Response:

This often depends on the journal’s style. It could be done by the copy-editor if the journal requires it. So we have left the text as is.

8. Line 76: “trans(inner)membrane” should read “integral inner membrane protein”.

Response:

We have corrected.

9. Line 80: "the DotL's C-terminus" should read "the C-terminus of DotL"

Response:

We have corrected.

10. Line 41: *Legionella* should be italicized.

Response:

We have corrected.

11. Line 124-125: "both N and C terminal" should read "both N- and C-terminal".

Response:

We have corrected.

12. Line 144: "While structure homology search" should read "While the structure homology search".

Response:

We have corrected.

13. Line 15-155: "27-30 amino-acids long, derived from the C-terminus of" should read "27-30 amino acids long, derived from the C-termini of".

Response:

We have corrected.

14. Line 178-179: remove "see location of the targeted residues in"

Response:

We feel this remark is important and therefore we have not removed it.

Reviewer #2

The manuscript by Meir et al presents very interesting and novel information regarding the way by which the Legionella pneumophila Dot/Icm complex recognizes a subset of its effectors using the DotM protein. The authors present the crystal structure of DotM which contains a large patch of positively charged amino acids residues. This large patch of positively charged residues was found to be important for the binding of the Glu-rich signal motif located at C-terminus of some effectors. This finding was validated using site directed mutagenesis of the DotM positively charged residues and by binding of the OSM synthetic signal as well as effectors C-terminus to DotM. In addition, the DotM mutants were introduced into the L. pneumophila genome and examined for their effect on intracellular growth and effector translocation using the CyaA translocation system. The DotM mutations were found to severely affect intracellular growth as well as the translocation of C-terminal

30 residues sequence of two *L. pneumophila* effectors. The manuscript is very well written and easy to follow and the *in vitro* experiments are well performed. However, the *in vivo* work requires few additions and controls in order to strengthen the conclusions made.

Major comments

Comment 1:

The translocation analysis using the DotM mutants M1 and M4 – the authors should examine if the DotM mutants are stable in-vivo. As indicated in lines 188-190, the authors crystalize the mutants and their structures were found to be virtually identical to the wild-type protein. However, the mutations might still affect the stability of the proteins in-vivo, or the ability to bind the other two components of the coupling complex DotL or DotN. A simple western using DotM specific antibody can address this point, or generating a genomic tagged version of DotM in case antibodies are not available.

Response:

Please see our response to comment 4 of reviewer 1 repeated here for convenience.

Using DotM antibodies (kindly provided by Professor Joseph Vogel from Washington University in Saint Louis), we monitored the stability and expression level of the DotM and DotM variant in our strains. These results are now reported in Figure S2A and demonstrate that the mutated proteins are as stable as wild-type and also expressed in equal quantities as wild-type. We have added on page 10: “Production of the DotM, M1 and M4 proteins in cells was monitored using anti-DotM antibodies and all wild-type and variant DotM proteins were shown to be produced in equal quantities (Figure S2A).”

Comment 2:

The translocation analysis of “IcmSW-dependent effectors” using wild-type DotM and the M1 and M4 mutants – the use of only the C-terminal 30 residues sequence of the effectors (Lem21 and LegC8) in this analysis, is problematic. As the authors indicated, the weak translocation of these effectors might result from the absence of internal IcmSW binding sites. Therefore, this analysis is not informative. However, it is very important to determine whether the DotM mutants affect the translocation of IcmSW-dependent effectors or not. The authors should examine full length constructs of these two effectors using the CyaA system in order to resolve this issue, since this point is important for the understanding of effector translocation by the Dot/Icm complex.

Response:

This experiment is also requested by Reviewer 1 (see comment 4). We have examined the full-length constructs of LegC8 using the CyaA system and have resolved this issue. We now present these results in Figure S2B and have amended the text accordingly. It now reads: “The Cya-LegC8_{Cter} and Cya-Lem21_{Cter} reporter fusions were translocated very weakly by all strains (Figure 6C) and there was not statistically-significant differences between strains. Weak translocation of these fusions is presumably due to the absence of internal IcmSW binding sites. Thus, to ensure that this is the case, full-length LegC8 was fused to the C-terminus of Cya (yielding a fusion termed Cya-LegC8) and translocation of Cya-LegC8 was monitored using the wild-type, M1 and M4 Legionella strains (Figure S2B). We observed higher levels of translocation, yet still no statistically-significant differences between wild-type and mutant strains, suggesting that weak translocation of Cya-LegC8_{Cter} is indeed due to the absence of an IcmSW-dependent translocation signal. Also, the fact that the full-length Cya-LegC8 fusion is translocated with the same efficiency in all Legionella strains indicates that the M1 and M4 mutations do not affect the IcmSW-dependent translocation function of the DotMLN complex.”

We did not carry out the experiment on Lem21, being in the opinion that checking one of the two (LegC8) is sufficient.

Minor comments

1. Fig. 6B – for clearly, please divide panel 6B into two separate panels one for J774 and one for *A. castellanii*.

Response:

We have separated the panel.

2. Conclusions, line 304, it is reasonable to assume that most of the dot/icm mutants will be found to be defective for translocation and not just the icmT mutant which was examined. Please rewrite the sentence, referring more to the positively charged residues of IcmT and less to its translocation phenotype.

Response:

We have removed the sentence.

Reviewer #3

DotM is a component of the T4BSS present in various pathogenic intracellular bacteria. In contrast to T4ASS, structural information on T4BSSs is limited. In this manuscript the authors present the crystal structure of DotM from Legionella pneumophila to a resolution of 1.8 Å, which together with DotL and DotN forms the so-called coupling complex in T4BSSs. The authors use a traditional X-ray crystallography approach for structure determination. The structure of DotM reveals charged surface regions that might facilitate binding of effector proteins to be exported via the membrane-spanning components of secretion system. To confirm this hypothesis the authors analyzed the interaction of DotM with peptides containing Glu-rich secretion motifs in vitro and the effect of DotM variants with disrupted charged surface regions in vivo.

Given the importance for virulence of a number of human pathogens, any structural information on T4BSSs is of interest for a broader audience. Although the structure of DotM in complex with other T4BSS components would greatly enhance this study in general, I understand that complex preparation, crystallization and structure determination would most likely take months to years, and is therefore outside of the scope of this manuscript.

The experiments presented in this study appear to have been carefully done. The manuscript is well written and conclusions should be understandable to non-specialists. The introduction section covers current literature on the topic appropriately. The method section is very clear and includes enough details to enable reproduction of the experiments. However, I feel that several points and questions need to be addressed to before this manuscript can be accepted for publication in Nature Communications.

Comment 1:

Is the observed interaction between DotM and the peptides containing secretion motifs sequence specific? Do peptides of identical length but with a random sequence of negatively charged residues bind to DotM, too? The authors should perform appropriate ITC experiments to investigate if the

interaction of effectors with DotM is solely based on charge or if the sequence motif found in Legionella effector proteins is necessary.

Response:

We respectfully disagree with the reviewer and feel that the request goes far beyond the scope of the study presented here. In this study, we set out to elucidate the role of DotM in effector recruitment and transport, a task which we have successfully completed. We agree that investigating the binding specificity would be of great interest, but this would require carrying out at least another 30 ITC experiments on 30 or more different peptides each mutated to Ala or other residues at each individual residue position, clearly a study which warrants a separate paper. We however have made a mutation at residue 145 of the CegC3 signal sequence, mutating this residue from E to A, and shown that binding is abrogated. This mutation alone is therefore indicative of sequence specificity. We have added on page 10: *“The observation that a single mutation abrogates binding suggests that positioning of Glu residues along the surface of DotM might be important and therefore that sequence specificity might play a role in acidic Glu-rich motif interactions with DotM.”*

Comment 2:

The authors note that attempts at introducing M2 mutations might have failed due to a destabilization of the DotMLN coupling complex, however, the ΔT4BS strain is viable. This should be further explained.

Response:

What's happening in the M2 strain *versus* what's happening in the ΔT4BS strain is radically different: in the former strain, we presumably have formation of an unstable DotMLN complex, while in the ΔT4BS strain, there is no DotMLN complex at all. The presence of an unstable DotMLN complex would stress processes at the membrane and therefore may lead to non-viability. However, this is entirely speculative, and we are in the opinion that speculative statements should be kept to the minimum. We have amended the text and now say: *“but attempts at introducing M2 failed for reasons that remain unclear but could be due to a destabilisation of the DotMLN coupling complex resulting in activation of processes that are stressful to the cell”*.

Comment 3:

Based on the presented results it cannot be excluded that effector translocation defects of DotM variants with disrupted charged surface patches are a result of a defective formation of the T4BSS complex itself. This needs to be excluded to strengthen the authors' hypothesis that the observed effect is indeed based on defective effector binding.

Response:

This comment was addressed by producing a full-length LegC8 fused to Cya (see response to comment 4 of reviewer 1 and comment 2 of reviewer 2). These experiments clearly demonstrate that the T4BSS is functional in DotM wild-type and variants *Legionella* strains.

Comment 4:

The presented model for the CegC3-interaction with DotM153 was manually chosen to match the obtained ITC data. It does not represent an independent result and therefore cannot be used to strengthen the proposed interaction between Glu-rich peptides and DotM. Although this is briefly mentioned in the materials and methods section my feeling is that it should be included in the main text as well.

Response:

We now say on Page 9 (change underlined): “*The biochemical, structural and mutational data described above were next used to produce an in silico model of CegC3-interaction with DotM153, using first the docking server CABS³², and subsequently refined using FlexPepDock^{33,34}, a ROSETTA-based server.*” We hope that adding “mutational” makes it very clear that we used all available data to model the peptide. We must also note that, in a new experiment, we have now validated the model by mutating E145 to Ala within the CegC3 sequence and showed that binding to DotM is abrogated (see Figure 4).

Comment 5:

The authors state that in solution DotM153 and DotM119 forms monomers and dimers, respectively. The asymmetric unit of DotM119 crystals contains 2 molecules. Do those molecules form a stable dimer that was observed in solution? The authors should clarify (or at least comment) on the physiological relevance of the different oligomeric states of DotM.

Response:

We are a little confused by this comment. DotM119 forms monomers in solution: it does not form stable dimers. The fact that DotM153 forms dimers in solution is irrelevant since the larger proteins (Dot119) forms monomers. So we don't believe we need to comment on the oligomeric state of DotM.

Comment 6:

I annotated a small number of typos in the attached pdf document.

Response:

We're immensely grateful to the reviewer for having corrected these typos and we have now incorporated the corrections into the text.

REVIEWERS' COMMENTS:

Reviewer #1 (Remarks to the Author):

I am happy with the author's thorough revisions, which address all my major concerns.

The structure of DotM and its role as a docking site for a specific class of effectors through electrostatic interactions are very exciting and provide valuable information about how effectors are targeted for translocation.

Reviewer #2 (Remarks to the Author):

The authors have answered my queries, I have no additional major comments.

The authors may consider the following minor comments:

1. Some of the proteins in the Dot/Icm system have both Dot and Icm designations. It would be convenient for some of the reader to include the Icm designation in the first time when the DotM, DotL and DotN proteins are mentioned.
2. Figure 5 – separating panel B into two panels is better in comparison to the previous version. But giving each of them a separate letter (B and C) would have been much better in comparison to referring to the left/right panel of Fig. 5B.
3. Figure 6 – the results of the full length LegC 8 effector are important and the authors should consider including them in the manuscript itself and not as a supplementary figure.

REVIEWERS' COMMENTS:

Reviewer #1 (Remarks to the Author):

I am happy with the author's thorough revisions, which address all my major concerns.

The structure of DotM and its role as a docking site for a specific class of effectors through electrostatic interactions are very exciting and provide valuable information about how effectors are targeted for translocation.

Response: no action needed

Reviewer #2 (Remarks to the Author):

The authors have answered my queries, I have no additional major comments.

The authors may consider the following minor comments:

1. Some of the proteins in the Dot/Icm system have both Dot and Icm designations. It would be convenient for some of the reader to include the Icm designation in the first time when the DotM, DotL and DotN proteins are mentioned.

Response: The corresponding Icm designations for DotM, L, and N have been added when these proteins are mentioned for the first time.

2. Figure 5 – separating panel B into two panels is better in comparison to the previous version. But giving each of them a separate letter (B and C) would have been much better in comparison to referring to the left/right panel of Fig. 5B.

Response: We do not believe that this suggestion would result in a clearer figure. We have therefore left it as is.

3. Figure 6 – the results of the full length LegC8 effector are important and the authors should consider including them in the manuscript itself and not as a supplementary figure.

Response: This is certainly an important control. However, moving into the main text would distract from the main message. Therefore, we do not believe it needs to be moved to the main text.